# Helical reconstruction of VP39 reveals principles for baculovirus nucleocapsid assembly

Friederike M. C. Benning [1,2], Simon Jenni [3], Coby Y. Garcia[1,4], Tran H. Nguyen [1], Xuewu Zhang [5,6] & Luke H. Chao [1,2] ✉

Baculoviruses are insect-infecting pathogens with wide applications as biological pesticides, in vitro protein production vehicles and gene therapy tools. Its cylindrical nucleocapsid, which encapsulates and protects the circular double-stranded viral DNA encoding proteins for viral replication and entry, is formed by the highly conserved major capsid protein VP39. The mechanism for VP39 assembly remains unknown. We use electron cryomicroscopy to determine a 3.2 Å helical reconstruction of an infectious nucleocapsid of *Autographa californica* multiple nucleopolyhedrovirus, revealing how dimers of VP39 assemble into a 14-stranded helical tube. We show that VP39 comprises a distinct protein fold conserved across baculoviruses, which includes a Zinc finger domain and a stabilizing intra-dimer sling. Analysis of sample polymorphism shows that VP39 assembles in several closely-related helical geometries. This VP39 reconstruction reveals general principles for baculoviral nucleocapsid assembly.

Baculoviruses are a family of arthropod-infecting, double-stranded DNA viruses. The biosafety properties and high target specificity of baculoviruses have led to their development as biological control agents, with growing importance due to the increase in insecticide resistance, compounded by climate disruption[1,2]. Baculoviruses are also well-established biotechnological platforms for heterologous protein expression[3], vaccine production[4], and therapeutic gene delivery[5–9]. Despite decades of study, the structural basis for baculovirus assembly has remained unknown.

The *Baculoviridae* family comprises four genera: alphabaculoviruses, betabaculoviruses, gammabaculoviruses, and deltabaculoviruses, which all share a set of 38 core genes[10]. Alphabaculoviruses are subdivided into group I and group II nucleopolyhedroviruses (NPV), which differ in their genomic content[11]. *Autographa californica* multiple nucleopolyhedrovirus (AcMNPV) is an extensively studied group I alphabaculovirus, and a model system for baculovirus molecular biology. The circular baculoviral genome ranges from 80 to 180 kb in

size and encodes for 90–180 genes[12,13]. Upon synthesis in the viral stroma of the host nucleus, the baculovirus genome is packaged into rod-shaped nucleocapsids of 20–60 nm in diameter and 200–400 nm in length[14–16]. A hallmark of baculoviruses is the presence of two morphologically and functionally distinct virion types, which depend on the stage of their biphasic life cycle. Occlusion-derived virions (ODV) drive the primary infection in epithelial cells of the larval midgut, initiating the release of budded virions (BV), which spread the infection to other cells in the host. Outside the host, a protein matrix protects nucleopolyhedroviral ODVs in occlusion bodies. While ODVs and BVs differ in the origins and composition of their bilayer envelopes, they both share a common nucleocapsid structure[14,17].

The 39 kDa protein VP39 is the most abundant component of the nucleocapsid, with homologs in alpha- and betabaculoviruses[10]. VP39 forms the capsid shell encasing the nucleocapsid core[18–20]. Deletion of VP39 in *Bombyx mori* NPV (BmNPV) and AcMNPV results in complete loss of budded virions, nucleocapsids, and AcMNPV replication[21,22].

[1]Department of Molecular Biology, Massachusetts General Hospital, Boston, MA 02114, USA. [2]Department of Genetics, Harvard Medical School, Boston, MA 02115, USA. [3]Department of Biological Chemistry and Molecular Pharmacology, Harvard Medical School, Boston, MA 02115, USA. [4]Harvard College, Cambridge, MA 02138, USA. [5]Department of Biophysics, University of Texas Southwestern Medical Center, Dallas, TX 75390, USA. [6]Department of Pharmacology, University of Texas Southwestern Medical Center, Dallas, TX 75390, USA. ✉e-mail: chao@molbio.mgh.harvard.edu

Mutation of glycine 276 to serine (G276S) in BmNPV VP39 results in fewer infectious budded viruses[23]. To date, the field has lacked high resolution understanding of baculovirus nucleocapsid assembly. Structural studies of baculoviral nucleocapsids have proven challenging due to their flexibility, heterogeneity, and fragility. Helical diffraction analysis of negatively-stained nucleocapsids of the betabaculovirus *Spodoptera litura* granulovirus (SlGV) suggested a 12-start helix with stacked rings parallel to the helical axis[15]. Two helical reconstructions of in vitro assembled *Helicoverpa armigera* NPV (HearNPV) at 14 Å and 21 Å resolution reported VP39 helical assemblies that varied in diameter[16].

Mature baculoviral nucleocapsids contain a distinct apical cap and a basal structure[24]. Based on these ultrastructural observations, the current model for baculoviral nucleocapsid packaging proposes that an ATP-driven motor injects the viral genome into empty, preformed capsids, as seen in other viruses with genome sizes larger than 20 kb[25–28]. First, VP39 assembles into empty capsids emanating from basal structures in the viral stroma, a region in the host cell nucleus[24]. Upon viral DNA replication in the viral stroma, the viral phosphatase 38 K dephosphorylates the viral protein P6.9, allowing it to condense the DNA into nucleoproteins[29]. Second, high concentrations of replicated DNA trigger its packaging into the preformed capsids, presumably through a protein channel at the apical cap[25]. An ATP-dependent packaging motor is hypothesized to pump DNA through the portal. Upon complete genome injection, a structural protein blocks the portal to form mature nucleocapsids[25]. Preformed capsids are observed to grow longer when they are not packaged with viral genome, however, it remains unknown how capsid length is regulated. Despite the identification of key components, major questions in nucleocapsid assembly, DNA packaging, and capsid maturation remain.

Interactions with the VP39 protein regulate key steps in the baculovirus lifecycle, including nucleocapsid-dependent transport. Retrograde transport of nucleocapsids to the nucleus occurs through actin-based motility, and VP39 has been observed to interact with actin and to be essential for cellular and nuclear actin polymerization[30,31]. Interactions of VP39 with the motor protein Kinesin-1 facilitate anterograde nucleocapsid transport from the nucleus to the cell periphery along microtubules[32,33]. In addition, VP39 has been shown to interact with several viral proteins such as the phosphatase 38 K, the DNA-binding protein P6.9, a transcriptional activator IE-2, and FP25, which regulates the BV to ODV ratio[29,34–39]. VP39 of Group I and Group II NPVs are relatively conserved, however, while substituted VP39 self-assembles into empty capsid structures, no infectious nucleocapsids were observed[40]. How the VP39 assembly templates cytoskeleton-dependent transport, or viral genome packaging has also remained unknown. The principles of VP39 assembly into capsids remain poorly understood, limiting molecular understanding of the baculovirus lifecycle and restricting rational engineering of capsid properties.

Here, we present a 3.2 Å-resolution electron cryomicroscopy (cryo-EM) structure of the AcMNPV VP39 protein and its assembly into helical nucleocapsids. We overcame the challenges associated with structural studies of baculoviral nucleocapsids by integrating cryo-EM technical sample preparation and computational solutions. Our helical reconstruction reveals that dimers of VP39 assemble into tubes of approximately 50 nm diameter and several hundred μm in length. We find that the VP39 monomer comprises a distinct mixed alpha/beta fold, which includes a putative Zinc finger domain facing the inside of the nucleocapsid. VP39 dimers are stabilized by wrapping a loop around a β hairpin of the adjacent monomer. Our analysis of the flexible subunit contacts provides insights into the assembly of nucleocapsids of varying diameters. Additionally, the AcMNPV nucleocapsid reconstruction reveals putative binding sites for the viral genome. We discuss the implications of these results for nucleocapsid assembly, genome packaging, viral trafficking, and infectivity.

## Results

### Helical reconstruction of the cylindrical central trunk of the VP39 nucleocapsid

We determined a helical reconstruction of the *Autographa californica* multiple nucleopolyhedrosis virus (AcMNPV) nucleocapsid (Fig. 1). Cryo-EM images were collected from secreted nucleocapsids purified from *Spodoptera frugiperda* Sf9 cells, which ranged up to several micrometers in length and had a diameter of approximately 40–55 nm (Fig. 1A). To maximize the number of intact nucleocapsids, we used a Sephacryl S-1000 SF resin for gel filtration and concentrated the sample using solvent absorption chambers, which yielded more intact tubes than centrifugal concentrators. Cryo-EM data collection was greatly facilitated by graphene-supported grids, which increased particle yield per micrograph by a factor of 5 and made this reconstruction possible[41]. With an initial data set collected from conventional Quantifoil grids (16,402 segments from 45,000 micrographs), we were only able to obtain a low-resolution reconstruction. We obtained 74,620 segments after manually picking tubes from 44,540 micrographs (Supplementary Fig. 1). 2D classification revealed significant variation in tube diameter between classes (ranging from ~36 to 53 nm;

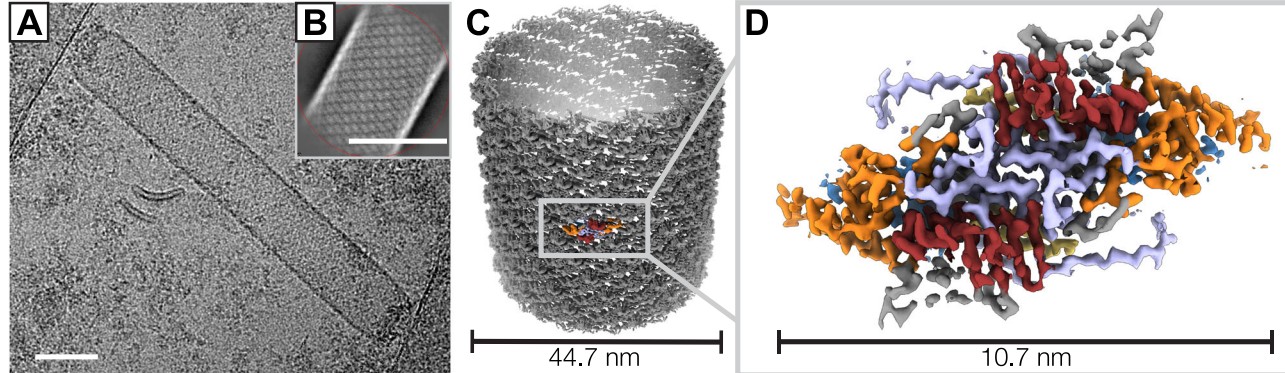

**Fig. 1 | Helical reconstruction of the AcMNPV nucleocapsid. A** A representative micrograph showing a vitrified AcMNPV nucleocapsid on a graphene-coated gold grid. Sigma contrast was set to 3 and the image was low-pass filtered to 20 Å. The scale bar corresponds to 50 nm. The micrograph was taken from two combined datasets of 45,996 movies. **B** A representative 2D class average image of nucleocapsid segments generated in cisTEM[69]. The scale bar corresponds to 50 nm.

**C** Helical reconstruction of the nucleocapsid at 3.6 Å resolution. **D** VP39 dimer volume at 3.2 Å resolution after local reconstruction and density averaging. Regions are color-coded: Zn-finger (residues 12–57, blue), antenna (residues 62–78, yellow), claw (residues 89–163, orange), glider (residues 176–225 and 301–304, red), lasso (residues 226–300, light purple).

Supplementary Fig. 2), suggesting heterogeneity in the sample. Initial helical symmetry parameters (rise: 43 Å, twist: −7.3°) were derived from 2D class averages of 27,777 segments selected from good-looking classes (see methods and Supplementary Fig. 1 for details)[42]. The segments within these classes had an outer diameter of 42.9 nm, and 3D maps reconstructed from a further selection of 4983 segments with D14 symmetry imposed revealed clear secondary structure features[43]. Fourier-Bessel indexing of the average power spectrum of these 4983 segments confirmed the initially determined helical symmetry parameters (Supplementary Fig. 3). After further refinement of segment alignment, helical symmetry parameters (final refined helical rise of 43.86 Å and helical twist of −7.16°), correction for beam-induced aberrations, and optimization of movie frame summation (see methods), we obtained a reconstruction with a resolution of 4.1 Å as judged by its Fourier shell correlation (FSC) with the final atomic model (see below) and applying a cutoff of 0.5 (Supplementary Fig. 4A, C).

In our reconstruction, individual VP39 subunits pack as dimers that assemble into 14 helical strands and together form the central cylindrical structure of the baculoviral nucleocapsid (Fig. 1C).

## Local reconstruction and modeling of the VP39 capsid protein

To improve the resolution for ab initio model building of the VP39 structure, we used a local reconstruction approach, focusing on aligning four adjacent VP39 dimers after symmetry expansion of the segment particle stack that we used for the helical reconstruction to 209,286 images (see methods). To improve the helical and local reconstructions, we then used the VP39 dimer in a supervised classification approach to identify segments that classified with the most abundant helical symmetry and corresponded to 0–2% of tube flattening. The resulting 19,012 segments were used for a final round of helical and local reconstruction (see methods and Supplementary Fig. 1).

After local alignment, we estimated the overall resolution of the four-dimer reconstruction to be approximately 3.2 Å (Fig. 1D, Supplementary Fig. 4B, D, Supplementary Fig. 5). The density map revealed unambiguous side-chain densities (Supplementary Fig. 6). We initially used ModelAngelo for model building[44], where we observed that—without providing an amino acid sequence—the program was able to output an almost complete trace of the VP39 structure with a high degree of correct amino acid assignments (64% identity, 77% similarity), confirming the visually assessed quality of the reconstruction. We completed the model by manual building in Coot[45]. The first 11 residues of the N terminus and the last 27 residues of the C terminus of VP39 were unresolved in our cryo-EM reconstruction and were not included in the model (Supplementary Table 1).

## Structure of the A. californica MNPV VP39 capsid protein

The helical reconstruction reveals that VP39 is assembled as dimers that form a compact repeat unit of 107 Å × 50 Å × 43 Å (Fig. 2A). Each dimer subunit comprises a mixed alpha/beta fold, with extensive interdigitation of elements with its partner in the dimer (Fig. 2B, D). Protein fold similarity searches of our determined VP39 monomeric unit, using the Dali Protein Structure Comparison Server[46] and Foldseek[47], did not result in significant hits from previously determined or predicted structures (Z score <5 for DALI), indicating the VP39 fold we describe here had not been previously observed experimentally.

The N terminus of the VP39 polypeptide chain folds into a Zinc finger (ZF) consisting of three short α helices, α1–α3 (residues 12–57; Fig. 2C). The VP39 ZF consists of a conserved C-x17-C-x12-C-x2-H motif (Supplementary Fig. 7). The Coulomb potential map reveals strong signal in the vicinity of cysteine 18, cysteine 36, cysteine 49, and histidine 52, which are positioned for tetrahedral coordination of a metal ion. Based on the nature and distances of the donor residues, which occur most frequently in Zinc coordination groups[48], we placed a Zinc

metal ion at the center of the coordination group. The ZF faces the luminal volume of the VP39 nucleocapsid assembly and is surrounded by positively charged residues, consistent with a role in binding viral DNA (Fig. 2E; Supplementary Figs. 7, 8).

Following the ZF, a short linker connects to a long β hairpin (consisting of β1 and β2), which we refer to as the antenna. An unstructured 10-residue long polypeptide segment links the antenna to the mixed alpha/beta claw, which cradles the ZF domain. A 25 Å-long finger (residues 19-36) of the ZF extends parallel to α helices α8-α10 of the claw and α helix α3 perpendicular to the finger region. A flexible 12-residue long stretch of the polypeptide chain leads from the claw to a distinct sub-region containing three α helices encircling a four-stranded β sheet. We refer to this section as the glider region. The glider region sits against the antenna and claw, completing the core VP39 domain. The glider β sheet comprises two central parallel β strands flanked by antiparallel β strands. A 75-residue long lasso-like extension from the glider β sheet forms a three-stranded antiparallel β sheet with strand β3 of the claw and wraps around the antenna domain of the adjacent monomer on the internal face of the tube. It then threads back into the glider domain as internal strand β9 (residues 226–300; Fig. 2D).

Residues 252–261 lie on a flexible loop that is exposed to the inside of the nucleocapsid. While there is density for the peptide backbone in this region, side chain density is not resolved.

## VP39 monomers wrap around their neighbor to form a dimeric helical repeat unit

The intra-dimer interface is stabilized by the 75-residue long lasso loop, parts of which protrude into the adjacent monomer of the repeat domain and wrap around its antenna domain (Fig. 2D). The dimer interface consists primarily of hydrophobic contacts between residues 225–291 of the lasso region of one monomer and residues 62–72 of the antenna of the adjacent monomer. An additional hydrogen bond is formed between aspartate 44 in the ZF region and tyrosine 288 in the lasso region. Further stabilizing interactions between the lasso loop and the antenna include salt bridges between lysine 273 and aspartate 70, as well as aspartate 282 and lysine 75. An intra-dimer lasso to lasso electrostatic connection occurs between arginine 225 and aspartate 245 (Supplementary Table 2). The interdigitated nature of the dimer interface together with the partial wrapping around the neighboring monomer gives the two monomers the characteristics of interlocking chains.

## Residues on flexible linkers connect helix repeat units

We refer to inter-dimer contacts, as opposed to intra-dimer contacts, for interactions between the VP39 dimeric helical repeat units. Using the PDBePISA tool[49], we identified four inter-dimer contacts, which are all primarily mediated by hydrogen bonding (Fig. 3, Supplementary Table 3). These inter-dimer subunit contacts contribute to nucleocapsid stability.

Lateral contacts stabilizing adjacent units in the nucleocapsid (Fig. 3, square symbol) are predicted to be the strongest inter-dimer interactions of the helical assembly (the theoretical total gain in solvation free energy upon interface formation is $\Delta G_{sol} = -13.8$ kcal/mol; obtained from analysis using PDBePISA and defined as the difference in total solvation energies of isolated and interfacing chains). These contacts are mediated by one pair of hydrophobic residues on two neighboring claw regions, which include the conserved cysteine 169. Residues involved reside on a flexible loop and on the outermost α helix (α10) of the claw. Notably, cysteine 169 is within 5.5 Å ($C_\beta$-$C_\beta$ distance) of another cysteine (Cys 132) of a laterally adjacent unit. Both cysteines were reduced even though the condition during purification was oxidizing (Supplementary Fig. 9).

The VP39 nucleocapsid is stabilized by two distinct types of interactions along the helical axis, referred to as axial contacts

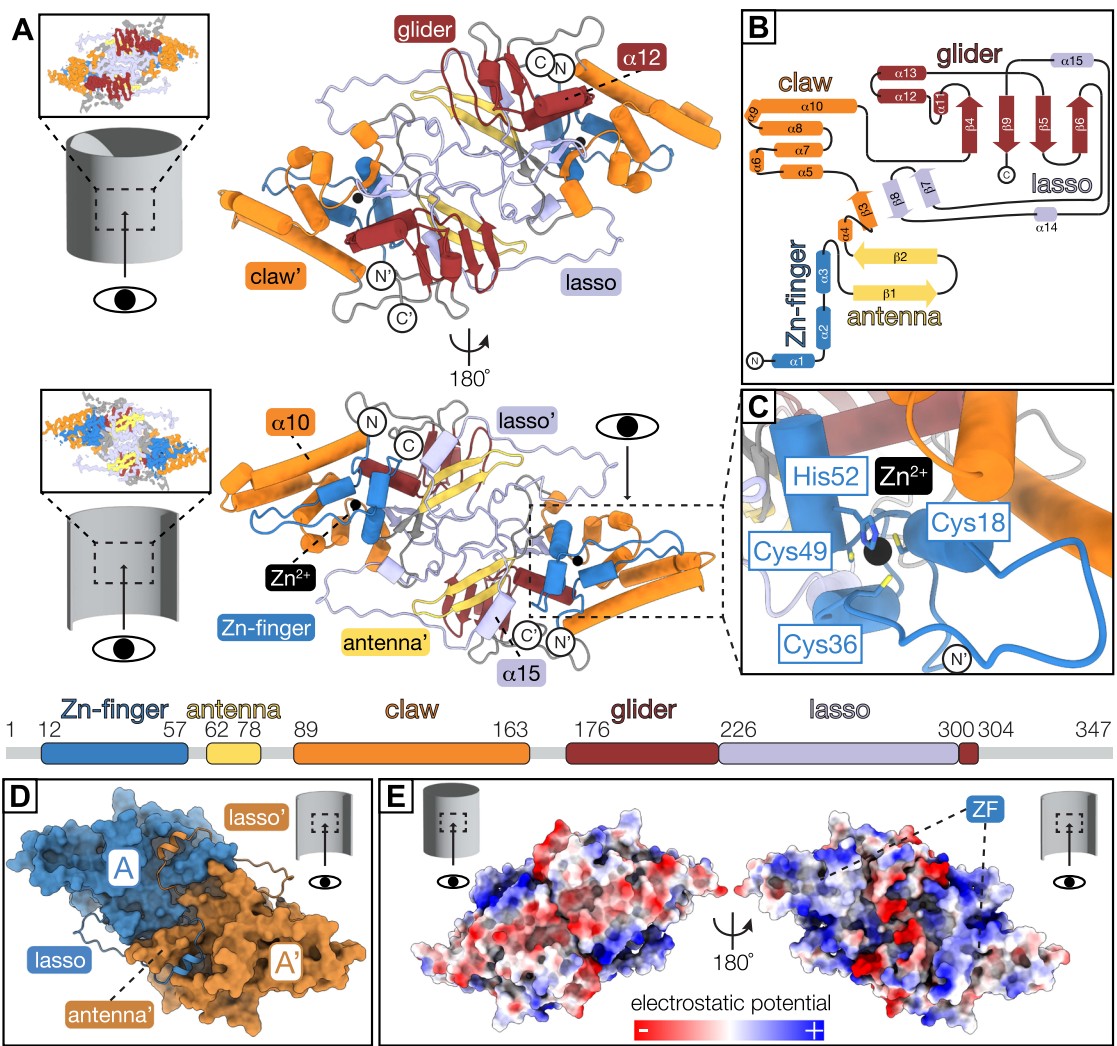

**Fig. 2 | The dimeric VP39 repeating unit comprises a distinct mixed alpha/beta protein fold. A** VP39 dimer model viewed from the capsid exterior (top) and luminal side (bottom). The linear fold architecture of the monomer is depicted below the models. Regions are color-coded: Zn-finger (residues 12–57, blue), antenna (residues 62–78, yellow), claw (residues 89–163, orange), glider (residues 176–225 and 301–304, red), lasso (residues 226–300, light purple), $Zn^{2+}$ (black sphere). Apostrophe indicates second monomer. **B** Secondary structure diagram of the VP39 monomer with regions colored according to linear scheme in (**A**). **C** Close-up view of the Zinc coordination site (more details in Supplementary Fig. 7). **D** Surface representation of the VP39 dimer model with monomers colored in blue (monomer A) and ocher (monomer **A'**). Residues 271–291 of the lasso region (cartoon representation) fold around the antenna region of the adjacent monomer. **E** Electrostatic surface potential of the VP39 dimer viewed from the exterior (left) and luminal (right) capsid side (more details in Supplementary Fig. 8). Electrostatic potential is colored from negative (red) to positive (blue). ZF Zn-finger region.

hereafter. Type-i axial contacts are exclusively hydrophobic and occur between residues of the claw and lasso regions with residues of the C terminus (A-D, A'-D', B-C, and B'-C'; $\Delta G_{sol} = -3.4$ kcal/mol; circled i symbol in Fig. 3). Type-ii axial contacts occur between monomers A' with D and B' with C ($\Delta G_{sol} = -6.2$ kcal/mol; circled ii symbol in Fig. 3). Type-ii axial contacts are mediated primarily by hydrophobic interactions of residues on the exposed, flexible portions of the lasso loops at the periphery of the subunit and a salt bridge, which links glutamate 275 and lysine 269.

Additionally, a hydrophobic contact stabilizes the axial-lateral interaction between the highly conserved glutamate 139 in the claw of monomer A' and the peptide backbone of phenylalanine 26 in the ZF region of monomer C ($\Delta G_{sol} = -5.0$ kcal/mol; hexagon symbol in Fig. 3).

### Conservation of the VP39 fold across all baculoviruses
To analyze the conservation of the VP39 fold and its interactions within the nucleocapsid across baculoviruses, we aligned 73 sequences of VP39 (the final alignment included sequences of 55

alphabaculoviruses, 15 betabaculoviruses, 2 gammabaculoviruses, and 1 deltabaculovirus) (Supplementary Figs. 10 and 11). In our alignments, the Zn-coordinating residues (cysteine 18, cysteine 36, cysteine 49 and histidine 52 in AcMNPV) in the ZF region are conserved in all 73 sequences. Moreover, several residues in helices in the claw and glider regions, which face the ZF coordination center ($\alpha$7, $\alpha$8, $\alpha$10, $\alpha$13), are highly conserved. Sequence conservation extends to several intra-dimer contacts as well as interactions between helical repeat units, including residues of the lasso region (residues tyrosine 250, leucine 266, isoleucine 268, valine 271, phenylalanine 274, glutamate 289), residue aspartate 44 in the ZF region, and cysteine 132, glutamate 139 and cysteine 169 in the claw (Supplementary Table 4). Additionally, a stretch of residues across the entire $\beta$ sheet in the glider region, which faces the outside of the tube, is conserved (Fig. 4A).

We generated protein structure predictions of VP39 dimers from six selected viruses using AlphaFold2[50,51]. To cover a wide range of baculoviral diversity, we selected sequences of a closely related alphabaculovirus (BmNPV), more distantly related alphabaculoviruses

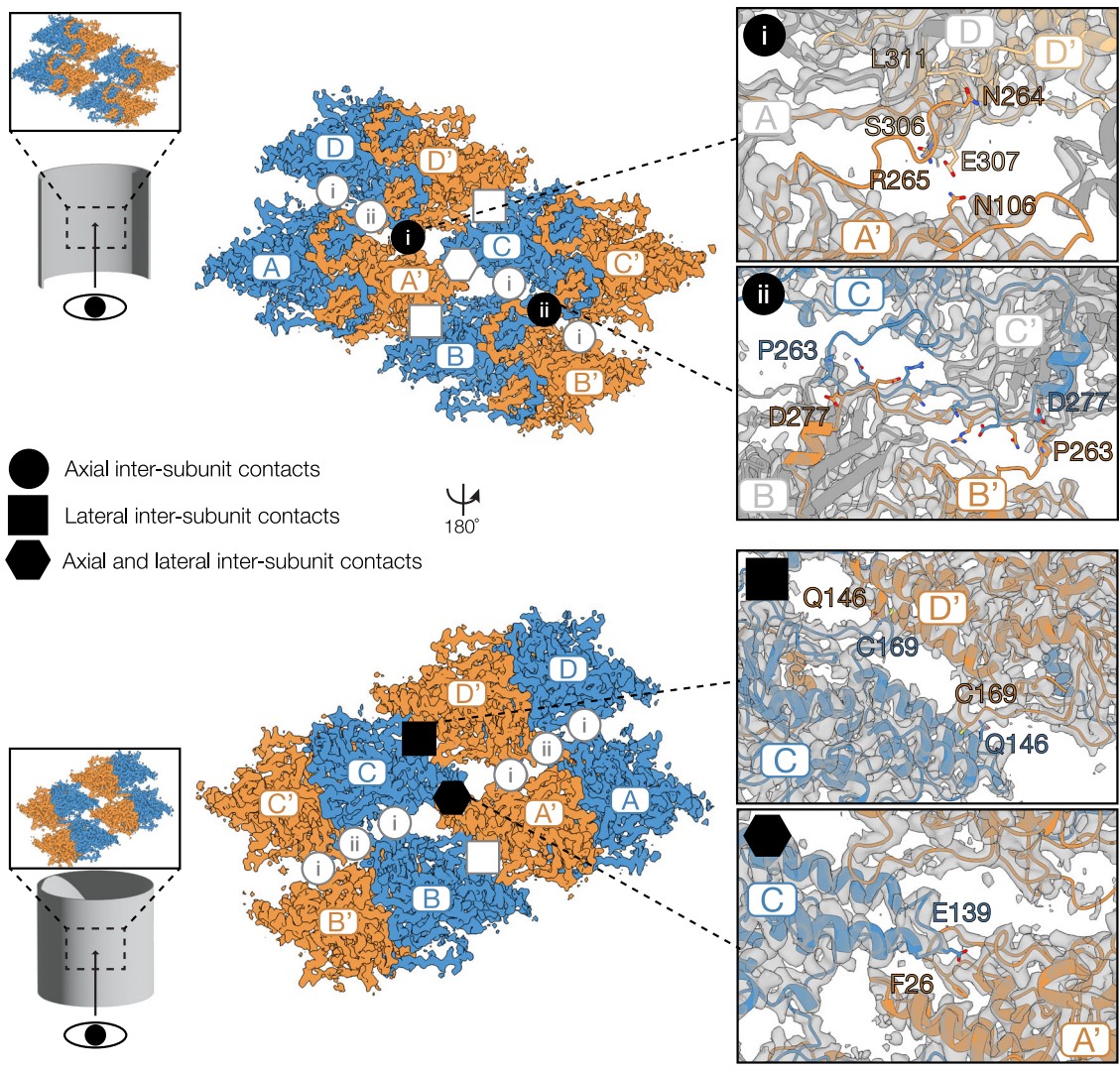

**Fig. 3 | Inter-dimer contacts of the VP39 assembly.** Four VP39 dimer repeat units and their interfaces viewed from the capsid lumen (top) and the capsid exterior (bottom). Monomers are labeled with letters (**A–D**) (blue) and A'-D' (ocher) for the dimeric partner monomer. The four types of inter-dimer interactions are highlighted as follows: axial contacts type-i and ii (circle with roman numeral), lateral contacts (square), axial-lateral contacts (hexagon). Close-ups of each type of contact, which are labeled with their respective symbol, show interface residues as sticks. Monomers, which are not involved in the respective type of contact depicted in the close-ups, are grayed out.

(HaNPV and SeNPV), and two betabaculoviruses (PxGV and SlGV; also called granuloviruses) for model prediction (Supplementary Fig. 12 for prediction confidence). Sequence-independent superposition of the predicted models onto our AcMNPV VP39 dimer structure reveals a high conservation of the VP39 dimer fold (Fig. 4B; 0.8–2.1 Å RMSD by residue). We limited the following model comparisons to regions of high prediction confidence (teal to blue on the local Distance Difference Test (lDDT) bar in Supplementary Fig. 12). In the ZF coordination center, which was predicted with high confidence, the four Zn-coordinating residues of all six models align with those in the reconstructed VP39 dimer (Fig. 4C, D). While all baculovirus dimers are predicted to share the same overall architecture, granulovirus models exhibit longer loops between α7 and α8 in the claw region compared to the alpha- and gammabaculovirus models (4 residues in alphabaculoviruses, 11 residues in PxGV and 32 residues in SlGV; Fig. 4E). Alphabaculoviruses also feature longer C termini than beta- and gammabaculoviruses, whose chains terminate directly after the β9 strand in the glider region. In the closely related alphabaculoviruses of *A. californica* and *B. mori*, residues 242–248 of the lasso region bulge out towards the outer surface of the tube, while it consists of a shorter

loop in other models (234–246 in HaNPV, 237–239 in SeNPV, 258–260 in PxGV, 254–256 in SlGV). While all models are predicted to have a lasso region, the prediction confidence for its central part is low. In contrast to the alpha- and betabaculovirus models, where the lasso encircles the antenna of the adjacent monomer, the lasso loop of the gammabaculovirus NsNPV is sandwiched between the antenna and the glider regions of the neighboring monomer (residues 232–300).

Our AcMNPV VP39 structure reveals that the luminal face of the dimer exhibits a net positive surface charge with a few distinct negative residues lining the dimer interface. The outward-facing side of the dimer features a band of positive charges in lateral direction. This charge pattern is conserved in the predicted alphabaculovirus structures and to a smaller extent in the beta- and gammabaculovirus models (Supplementary Fig. 13).

The multiple sequence alignment shows that residues involved in lateral, type-ii, and axial-lateral contacts are conserved across all baculoviruses. In particular, residues cysteine 169 and valine 271 are present in all baculoviruses, and glutamate 139 is present as a glutamate or aspartate. The candidate disulfide-partner cysteine 132 is conserved in all baculoviruses (Supplementary Fig. 10). In contrast,

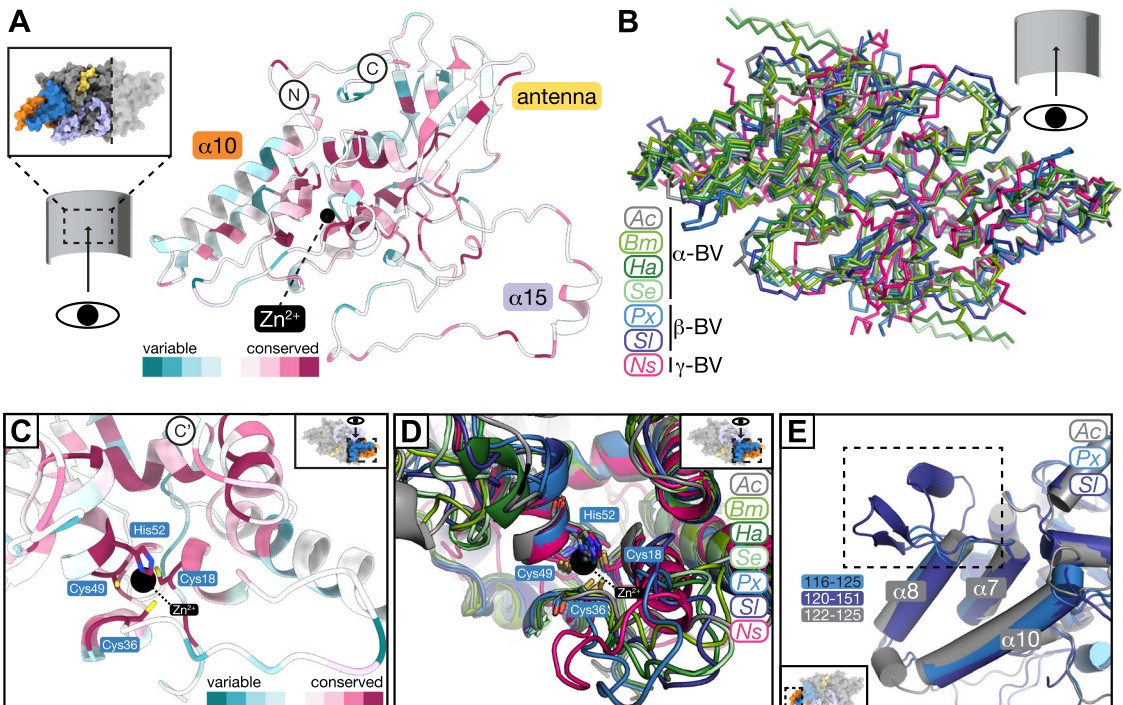

**Fig. 4 | Sequence and fold conservation of baculoviral VP39. A** Amino acid sequence conservation of VP39 across 73 sequences mapped on the VP39 monomer structure (viewed from the inside of the nucleocapsid) using the ConSurf webserver[86,87]. Variable residues are colored from light to dark teal with darker colors representing stronger sequence variability in sequence alignments. Light red to dark red represent conserved residues from less to high conservation. **B** Model predictions of baculoviral VP39 dimers superposed onto *A. californica* (*Ac*; gray) VP39 dimer structure. Predicted VP39 structures using AlphaFold2 (see methods) from the following viruses were superimposed: *B. mori* NPV (*Bm*; medium green; 1.5 Å RMSD by residue), *H. armigera* NPV (*Ha*; dark green; 1.1 Å RMSD by residue), *S. exigua* NPV (*Se*; light green, 1.0 Å RMSD by residue), *P. xylostella* granulovirus (GV)

(*Px*; light blue; 1.1 Å RMSD by residue), *S. litura* GV (*Sl*; dark blue; 1.1 Å RMSD by residue), *N. sertifer* NPV (*Ns*; pink; 5.0 Å RMSD by residue). α-BV: alphabaculovirus, β-BV: betabaculovirus; γ-BV: gammabaculovirus. **C** Close-up of the ZF region of AcMNPV VP39 with Zn-coordinating residues depicted and color-coded by sequence conservation using ConSurf. **D** Close-up of the ZF region of AcMNPV VP39 with superposed predicted models for six baculoviral VP39 dimers (see (**B**) for details and color code) with Zn-coordinating residues shown. **E** Close-up of the inward-facing claw region of AcMNPV VP39 with superposed predicted models of VP39 of betabaculoviral *P. xylostella* GV (*Px*; yellow) and *S. litura* GV (see (**B**) and methods for details on prediction and superposition).

residues asparagine 264 and leucine 266, which are involved in type-i interfaces, and the axial-lateral contact residue phenylalanine 26 are conserved exclusively in alphabaculoviruses but not in other baculoviruses (Supplementary Fig. 10; Supplementary Table 4).

Superposition of the predicted baculoviral dimer structures shows that both salt bridges that stabilize the intra-dimer interface (lysine 273 and aspartate 70; aspartate 282 and lysine 75) are conserved in alphabaculoviruses. In BmNPV, lysine 273, aspartate 70, aspartate 282, and lysine 75 structurally align with the identical residues in AcMNPV. In HaNPV, structural alignment of arginine 261 with lysine 273 and glutamate 68 with aspartate 70 suggest that this salt bridge may be conserved. In SeNPV, lysine 274, glutamate 69, glutamate 277, and arginine 75, which faces in the opposite direction, are structurally aligning with the salt bridge residues in AcMNPV.

### Nucleocapsid polymorphism

Heterogeneity in helical assemblies of biomolecules has been observed in several cases when analyzed by cryo-EM[52–55]. Possible reasons for the observed heterogeneity were variations in the helical packing of subunits, and artifacts introduced during cryo-EM sample preparation, such as deformation or flattening of assembled tubes during vitrification. We observed nucleocapsids of varying diameters on micrographs and the distribution of the measured diameter of all extracted segments is shown in Supplementary Fig. 2A, suggesting that not all nucleocapsids had the same helical symmetry as we applied during the reconstruction of a subset of segments. This heterogeneity

was consistent with differences distinguishable by 2D and 3D classification.

To further investigate the degree of polymorphism in our observed nucleocapsids, we applied a supervised classification approach to sort segments into classes with different helical geometries and/or potential tube flattening induced by sample preparation. The helical geometry of an AcMNPV nucleocapsid can be described by wrapping a 2D crystal lattice into a tube without creating a seam (Fig. 5A). This wrapping can be done by choosing different wrapping vectors, whose components are defined by the numbers $n_1$ and $n_2$ of unit cell vectors of the 2D lattice ($n_1\mathbf{a}$, $n_2\mathbf{b}$). Each wrapping vector leads to a tube with distinct helical geometry. If $n_1$ and $n_2$ are integer times of each other, the helical assembly has rotational symmetry, e.g., such as in our helical reconstruction (above), which is described by a wrapping vector with $n_1$ and $n_2$ of both 14 and consequently has C14 symmetry (or D14 if the dyad of the VP39 dimer is taken into account). We therefore created 144 polymorphic 3D reference structures, each of which with a distinct helical symmetry, for supervised 3D classification (Fig. 5B–D). By choosing different wrapping vectors, but maintaining the geometry of the 2D lattice, the local inter subunit contacts between VP39 dimers (Fig. 3) would not be substantially altered. In addition, we applied tube flattening (10 per reference), thus yielding a set of 1440 3D reference structures for supervised classification (see methods, Supplementary Figs. 14 and 15 for details).

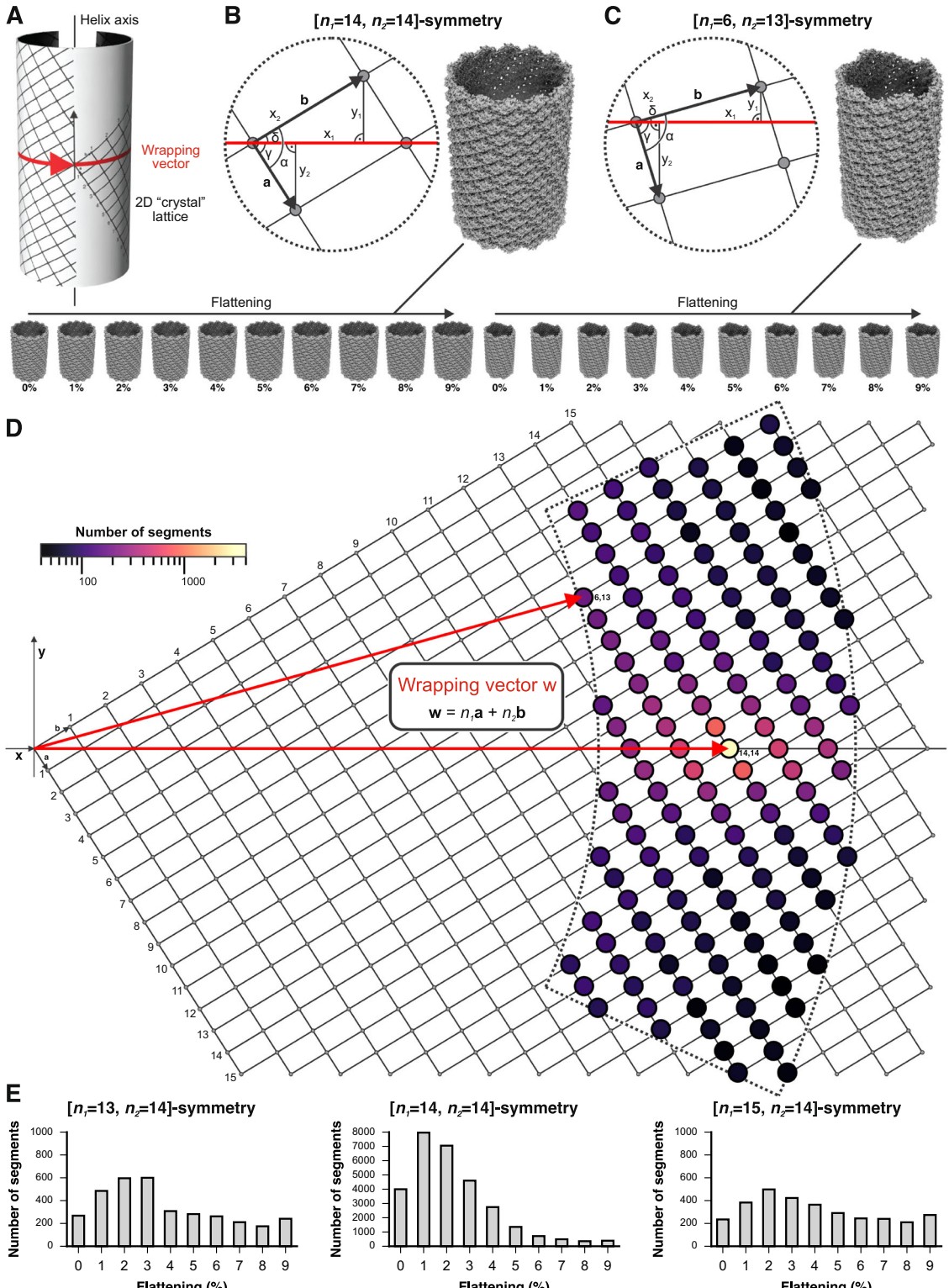

**Fig. 5 | Heterogeneity of the AcMNPV capsid. A** Formal description of helical symmetry by wrapping a 2D crystal lattice on a cylinder. A wrapping vector, shown in red, points from the origin of the lattice to a particular lattice point and lies in the equator of the helix. **B, C** Calculation of helical parameters from the lattice and wrapping vectors (see also methods) and preparation of 3D references for supervised classification. If $n_1$ and $n_2$ are integer times of each other, as with $[n_1 = 14, n_2 = 14]$-symmetry shown in (**B**), the helix has rotational symmetry. If $n_1$ and $n_2$ are not related by an integer value, as with $[n_1 = 6, n_2 = 13]$-symmetry shown in (**C**), there is no rotational symmetry. 3D references with increasing degree of flattening are shown at the bottom for the two helical symmetries. **D** Supervised classification results mapped on the 2D crystal lattice. The sector delineated by dashed lines includes the lattice points with corresponding different helical symmetries for which we prepared 3D references. Two wrapping vectors for the symmetries shown in (**B, C**), respectively, are shown in red. The distribution of VP39 capsid geometries is color-coded by prevalence from low (black) to high (yellow). **E** Histograms for three capsid geometries with substantial prevalence reveal the degree of tube flattening. Source data are provided as a Source Data file.

After global alignment (C1, no symmetry imposed) of 72,013 extracted segments to the 1440 3D reference structures and assigning each segment to the corresponding 3D reference which gave the highest score, we observed that 40% of the segments were part of capsids that assembled with a helical symmetry of $n_1 = 14$ and $n_2 = 14$, which is the symmetry of our helical reconstruction (Fig. 5D). Most of the remaining segments partitioned to helical symmetries with wrapping vectors in the vicinity of the [$n_1 = 14$, $n_2 = 14$]-helical geometry. A closer inspection of the segments belonging to tubes with [$n_1 = 14$, $n_2 = 14$]-symmetry revealed that only approximately 14% of these segments aligned best to the non-flattened 3D reference. Different degrees of flattening were observed for the remaining [$n_1 = 14$, $n_2 = 14$]-segments, and classes with different helical symmetries showed comparable flattening distributions (Fig. 5E), suggesting that this type of tube distortion did indeed occur during sample preparation and accounts, in addition to different helical geometries, for the observed distribution of tube diameters in the cryo-EM images and the observed heterogeneity of 2D class averages (Supplementary Fig. 2). The imposed symmetries for segments partitioning into 3D references are consistent when comparing the direct fit of the average power spectra of the segments and the calculated power spectra from the back-projection of the corresponding 3D references (Supplementary Fig. 14, Supplementary Movies 3 and 4). Given the caveats associated with analyzing Fourier spectra[56,57], interpretable maps would be the best way to verify correct helical symmetry assignments. However, while we calculated reconstructions for segments of two 0–2% flattened classes (Supplementary Fig. 16), we did not have a sufficiently high number of segments to yield an interpretable map to unambiguously prove the correct helical symmetry assignments for symmetries other than our [$n_1 = 14$, $n_2 = 14$]-reconstruction (Supplementary Table 5).

## Discussion

Baculoviruses have been used since the 1980s as gene vectors in academic and industrial research, and as pesticides against insect crop infestations. The rod-shaped nucleocapsid protects the large double-stranded viral DNA. While earlier work identified VP39 as the major capsid protein, how the baculoviral nucleocapsid assembly is formed by VP39 subunits has remained unknown until now[18–20]. Here, we present a high-resolution helical reconstruction by cryo-EM of the *A. californica* multiple nucleopolyhedrovirus (AcMNPV) nucleocapsid central trunk, which we purified from infected *Spodoptera frugiperda* Sf9 cells. At 3.2 Å overall resolution, side chains were clearly identifiable, which allowed us to build a high-quality model of the VP39 protein within the local reconstructed cryo-EM volume (Fig. 1).

Our reconstruction allows an in-depth analysis of the VP39 assembly. We find that VP39 forms a dimeric repeat unit that assembles predominantly into a 14-strand helix. The VP39 dimer subunit adopts a distinct mixed alpha/beta fold (Fig. 2), which was not recognized by the DALI structural comparison server and Foldseek. The fold includes a Zn-finger region with a $Zn^{2+}$ ion coordinated by a conserved CCCH motif, which is facing the capsid lumen. Moreover, it is surrounded by basic residues, priming this region as a possible binding pocket for the viral DNA. Notably, a 75-residue long lasso loop embraces the antenna region of the adjacent VP39 monomer in each repeat unit, providing significant additional dimer stability and thereby forming interlocking dimer pairs.

A mutagenesis study of BmNPV VP39 identified the conserved glycine 276 residue as important for proper nucleocapsid assembly[23]. Morphologically aberrant nucleocapsids have been previously observed for a G276S mutant[23]. Our reconstruction reveals that glycine 276 is nestled in between several axial assembly contacts facing the nucleocapsid lumen and introduces a significant kink in the lasso region (Fig. 3, Supplementary Fig. 17). This explains how glycine 276 plays an important role in maintaining the integrity of the lasso loop and associated inter-dimer contacts.

Actin-based motility has been shown to accelerate nuclear baculoviral nucleocapsid transit and egress[58]. Published work identified residues 192–286 in BmNPV VP39 to be required for nuclear actin polymerization[30]. Our reconstruction shows that these residues form a spiraling pattern along the exterior side of the capsid shaft (Supplementary Fig. 18). AcMNPV shares 97% sequence identity with BmNPV for this putative actin-binding site.

To further investigate observed variations in nucleocapsid diameter, we performed a supervised 3D classification analysis of our dataset (Fig. 5). The analysis allowed us to determine to which extent VP39 assembles into nucleocapsids with different helical geometries (while maintaining essentially identical inter-dimer contacts), and to which extent cryo-EM sample preparation distorted the nucleocapsid tubes[14]. We found that 40% of the tubes in our dataset assembled with [$n_1 = 14$, $n_2 = 14$]-helical symmetry. Most of the remaining segments belonged to tubes of either C15 or C13 symmetry, which correspond to occasional insertions or deletions of single subunits in the helical plane. These differences in helical symmetries generate nucleocapsids that match the published range of AcMNPV diameters. Our analysis furthermore indicated that most of the tubes were flattened in our preparation. While we cannot unambiguously identify the source of flattening in our sample, it is noteworthy that tube flattening has been observed for cilia in published work[59], with one study describing tube compression parallel to the ice plane on the EM grid[59,60].

In contrast to more rigid helical assemblies, the flexibility and elasticity of filaments often present challenges for high-resolution structural analysis[61]. Filament conformational heterogeneity or variation in assembly state can pose particular challenges for identifying the correct helical symmetry at low resolution. Frequently, near-atomic resolution is required to unambiguously distinguish the correct symmetry parameters[62]. For our VP39 nucleocapsids, the fragile nature of the sample and the variety of observed nucleocapsid diameters likely hampered previous attempts at structure determination. We overcame these limitations by combining the use of graphene-coated EM-grids and solvent absorption chambers for sample concentration. Careful classification to select a subset of heterogenous segments and focused local reconstruction approaches provided critical steps to obtain our near-atomic resolution final maps.

Earlier cryo-EM analysis of AcMNPV nucleocapsid ultrastructure in budded vesicles reported two distinct AcMNPV nucleocapsid morphologies, which differ in diameter[14]. Similarly, low-resolution cryo-EM reconstructions of recombinantly expressed and self-assembled HaNPV VP39 nucleocapsids reveal two capsid morphologies, which differ in diameter (90% self-assemble into the narrower one)[16]. Each study compares the two nucleocapsid morphologies to relaxed and compressed springs, thereby proposing that the nucleocapsid may expand or compress in accommodating the viral genome. We find that most of the residues that form inter-dimer contacts are on flexible loops, which may facilitate longitudinal capsid stretching and play a role in pushing the viral genome out of the nucleocapsid.

Our results show that the Zn-coordinating residues and several inter- and intra-dimer contacts of the VP39 dimer are conserved across alpha-, beta- and gammabaculoviruses. Based on structure predictions, we reveal that alpha- and betabaculovirural VP39 shares the same dimeric protein fold and general surface charge patterns. The charge pattern across the luminal face of the nucleocapsid suggests a surface primed for interacting with multiple strands of packaged DNA (Supplementary Fig. 8). While the low-confidence predicted lasso loop in gammabaculoviral VP39 is inserted between the antenna and glider regions of the adjacent monomer and introduces stronger bending of the VP39 dimer towards the capsid inside, all other regions share the same protein fold as alpha- and betabaculoviruses. The differences in tube diameter for different baculoviral species are likely due to different inter-dimer contacts. Based on the conserved fold and overall architecture, our reconstruction reveals general principles for

baculoviral nucleocapsid assembly. Future studies will explain outstanding questions including how the capsid is capped during maturation and how nucleocapsid interactions with cytoskeletal elements regulate the viral lifecycle.

## Methods

### Nucleocapsid expression and purification

*Spodoptera frugiperda* Sf9 cells (catalog number 12659017, Thermo Fisher Scientific, MA, USA) in SF900 III SFM medium (Thermo Fisher Scientific, MA, USA) were transfected with purified bacmid from DH10EMbacY cells (Intact Genomics, MO, USA), carrying the unrelated decahistidine- or FLAG-tagged human constructs, and FuGENE 6 transfection reagent (Promega, WI, USA), propagated, and stored as baculovirus infected insect cell stocks (BIICs)[63]. Nucleocapsids were harvested from Sf9 cells 96 h after incubation with BIICs by centrifugation at 1000 x g for 20 min and lysed in lysis buffer (50 mM HEPES, 0.25 M NaCl, 40 mM imidazole, 5% glycerol, 5 mM EDTA, pH 7.5) by sonication using a Branson 450 Digital Sonifier (Marshall Scientific, NH, USA) with 10 s pulses at 50% amplitude for 12 min. The lysate was cleared by ultracentrifugation at 142,000 x g and the pellet solubilized in lysis buffer with 1% (w/v) lauryl maltose neopentyl glycol (LMNG), followed by homogenization using a Polytron PT 1200 E homogenizer (Kinematica, Switzerland), sonication and incubation for 1 h at 4 °C. Solubilized protein was cleared by ultracentrifugation at 46,400 x g and purified by metal affinity chromatography using a Nickel-charged EconoFit Nuvia IMAC column (Bio-Rad, CA, USA). Protein was eluted in 20 mM HEPES, 50 mM NaCl, 5% glycerol, 1 mM EDTA, 0.02% (w/v) LMNG, pH 7.5 with a linear gradient to 1 M imidazole. Lysate containing FLAG-tagged target protein was purified using ANTI-FLAG M2 Affinity Gel resin (Sigma-Aldrich, MO, USA) and eluted with 3x FLAG peptide (APEx-BIO, TX, USA). Following concentration by centrifugation (Amicon Ultra, Millipore Sigma, MA, USA) the eluate was buffer exchanged into 20 mM Triethanolamine, 50 mM NaCl, 5% glycerol, 1 mM EDTA, 0.02% LMNG with a HiPrep 26/10 Desalting column (GE Healthcare, IL, USA). It was further purified by anion exchange chromatography using HiTrap Q HP resin (Cytiva, MA, USA) by a linear gradient to 1 M NaCl. The sample was concentrated to 0.5 ml and purified by size exclusion chromatography using Sephacryl S-1000 SF resin (Cytiva, MA, USA). Purified nucleocapsid tubes were concentrated using Vivapore solvent absorption chambers (Sartorius, Germany) and subsequently used for further processing. The presence of tubes was confirmed by negative stain transmission electron microscopy (TEM).

### Negative stain TEM

Glow-discharged 200 mesh FCF200-CU-SB carbon-coated copper grids (Electron Microscopy Sciences, PA, USA) were prepared by adsorbing 3.5 μl of the sample for 45 s, followed by three washes with water before staining three times for 10 s each in 1.25% uranyl formate. The grids were imaged either on a FEI Morgagni microscope operated at 80 keV equipped with an AMT Nanosprint5 camera (Brandeis Electron Microscopy Facility, Brandeis University, Waltham, MA, USA) or on a Phillips CM10 operated at 100 keV equipped with a Gatan Ultra-Scan 894 CCD camera (Molecular Electron Microscopy Suite, Harvard Medical School, Boston, MA, USA).

### Graphene grid preparation for single-particle cryo-EM

Graphene grids were prepared as described by ref. 41. Graphene on copper foil (Graphene Supermarket, NY, USA) was coated with methyl methacrylate MMA(8.5)MAA EL 6 (Kayaku Advanced Materials, MA, USA) using a tabletop centrifuge converted into a spin coater at the speed of 94 x g for 1 min. Backside graphene was removed by glow discharge at 30 mA for 30 s using a PELCO easiGlow system (Ted Pella, CA, USA). A MMA/graphene bilayer was created by etching off the

copper layer in 1 M ammonium persulfate (Sigma-Aldrich, MO, USA) for 20–30 min, followed by washing the bilayer in water. MMA/graphene was applied onto 300 mesh gold-coated Quantifoil R 0.6/1 copper grids (Electron Microscopy Sciences, PA, USA) by using the grids to scoop out the bilayer, followed by air-drying the MMA/graphene-supported grids. The graphene was annealed to the grids by incubating at 130 °C for 20 min. After cooling the grids to room temperature, the MMA layer was removed by two incubation steps in acetone for 30 min each, followed by incubation in 2-propanol for 20 min. Grids were air-dried, followed by incubation at 130 °C for 20 min. Graphene-coated grids were either used directly or vacuum-sealed for storage.

### Specimen preparation and cryo-EM data acquisition

Immediately prior to sample application, graphene-grids were made hydrophobic by exposure to UV/ozone for 10 min using a ProCleaner (Bioforce Nanosciences, VA, USA). 3.5 μl of sample were applied to the graphene-side of the grids, incubated for 30 s, followed by blotting for 6 or 8 s using a Vitrobot Mark IV at room temperature, 100% humidity, with a blotting force of +15, and vitrified in liquid ethane cooled to liquid nitrogen temperature. Grids were imaged using a Titan Krios (Thermo Fisher Scientific, MA, USA) microscope operated at 300 keV and equipped with a K3 direct electron detector and a GIF BioQuantum energy filter (Gatan, CA, USA). Two data sets of 19,883 and 26,113 movies, respectively, were collected using SerialEM version 3.8.5 at a nominal magnification of 105,000x with a pixel size of 0.825 Å and a defocus range of −0.4 to −1.9 μm. Movies were acquired with 2.7 s exposure time, fractionated into 52 frames and 13.88 e$^-$/pixel/s (total dose of 55.06 e$^-$/Å$^2$) for data set 1 and 13.683 e$^-$/pixels/s (total dose of 54.28 e$^-$/Å$^2$) for data set 2.

### Cryo-EM data processing and initial helical reconstruction

We collected two large data sets for this reconstruction (see Supplementary Fig. 1 for flow-chart). A first data set of 19,883 movies was corrected for beam-induced motion using the UCSF MotionCor2[64] program and the contrast transfer function (CTF) was estimated with CTFFIND4.1[65]. Micrographs with an estimated resolution worse than 10 Å (as judged by the quality of the fit of observed Thon rings) were removed. In the remaining 19,145 micrographs, we manually defined the start-end coordinates of observed tubes in Relion 4.0.1 and extracted 27,706 segments with the following parameters: 500 Å tube diameter, number of asymmetrical units: 1, helical rise: 44.2 Å, box size: 228 pixels with 3.3 Å/px (corresponding to four-times binned data)[43,66,67]. The helical symmetry parameters, which were used for segment extraction, were obtained by Fourier-Bessel analysis of 2D class averages of an initial data set of AcMNPV VP39 nucleocapsids on conventional Quantifoil grids (as opposed to final data collected on graphene-coated gold grids).

Following six rounds of 2D classification in Relion, we generated an initial volume from 4512 segments using relion_helix_inimodel2d[68]. Class averages from 2D classification in cisTEM (v. 1.0.0[69]) showed more detailed subunit features than those produced with Relion. We therefore subjected the 27,706 initially picked segments to three rounds of 2D classification in cisTEM v. 1.0.0[69], selected class averages that showed detailed subunit features and tubes of the same diameter, and generated a helical reconstruction from 9344 selected segments and the initial volume in CryoSPARC (twist: 7° with a search range of −15° to 15°, rise: 43.8 Å with a search range of 20–60 Å, C15, non-uniform refinement option)[70,71]. The resulting map was used as an input model for helical reconstruction in CryoSPARC with 8288 segments from two rounds of 2D classification of the initial 27,706 segments in cisTEM (twist: 7° with a search range of −15° to 15°, rise: 44 Å with a search range of 20–60 Å, C14, non-uniform refinement option). Finer selection in these and subsequent rounds of 2D classification, where we focused exclusively on class averages with C14 rotational

symmetry, led to switching from C15 to C14 symmetry for helical reconstruction.

The initial collection of 19,883 movies proved to be insufficient for calculating a 3D reconstruction with identifiable secondary structure elements. We therefore had to increase the number of segments and collected a second set of 26,113 movies. These movies were pre-processed in Relion as described above for data set 1, yielding 46,914 segments, which were extracted with the following parameters: 500 Å tube diameter, number of asymmetrical units: 1, helical rise: 44.2 Å, box size: 456 pixels with 1.65 Å/px (corresponding to two-times binned data). Following two rounds of 2D classification in cisTEM, 19,823 segments were combined with 7954 segments from data set 1 (re-extracted with a 456 px box at 1.65 Å/px and subjected to two rounds of 2D classification in cisTEM). We determined initial helical symmetry parameters by indexing the power spectra of these class averages using PyHI (Python v. 3.7)[42]. The 27,777 segments were used for helical reconstruction using the above mentioned C14-volume and applying D14-symmetry in CryoSPARC (twist: 7.5° with a search range of −15°–15°, rise: 44 Å with a search range of 20–60 Å, D14, non-uniform refinement option). We used CryoSPARC for helical recon-struction of the two-times binned data because secondary structures in maps generated with Relion using the same input were slightly flattened. Following one round of 3D classification (10 classes), in which we selected two classes with similar tube diameter and refined each class separately using the helical refinement option in CryoSPARC (twist: 7.5° with a search range of −15°–15°, rise: 44 Å with a search range of 20–60 Å, D14, non-uniform refinement option), a final helical refinement of the two combined classes (4983 segments, twist: 7.5° with a search range of −15°–15°, rise: 44 Å with a search range of 20–60 Å, D14, non-uniform refinement option), resulted in a map at 4.1 Å resolution (0.143 FSC at 4.1 Å using half maps, 0.5 FSC at 4.4 Å using map and final model [see below]; Supplementary Fig. 4A, C). The correctness of the imposed helical symmetry during reconstruction was supported by observation of resolved secondary structures (α helices and β strands) in this map.

### Refined helical reconstruction

One limitation of segment selection after 2D classification in cisTEM was the program's inability to carry forward all particle metadata, such as the extraction coordinates and name of the original micrographs. We therefore mapped back the 4983 segments of the initial helical reconstruction to the original segment particle stack from the combined two data sets. We did this by converting each image into a Python NumPy array[72], calculating an MD5 checksum, and mapping identical checksums between the two stacks using Python dictionaries. We then re-extracted with relion_preprocess the selected 4983 seg-ments from the original micrographs with the following parameters: box size: 912 pixels with 0.825 Å/px, background radius: 400 pixels (corresponding to unbinned data) (Supplementary Fig. 1).

We prepared a 3D mask from the initial helical reconstruction with relion_mask_create (--lowpass 4.2 --helix --z_percentage 0.6 --extend_i-nimask 10 --width_soft_edge 10). We used cisTEM for particle align-ment when working with large box sizes (912 cubic voxels) because of its speed and limitations due to memory requirements when using other software packages for alignment. To obtain a refined helical reconstruction from the 4983 segments, we used the following pro-tocol: (i) Local alignment with C1 symmetry of each segment to the current 3D reference (symmetrized and masked) using refine3d from cisTEM[69]. (ii) Calculation of 3D reconstructions with C1 symmetry (full and half maps) using relion_reconstruct. (iii) Refinement of the helical symmetry parameters in the C1 reconstructed maps using relion_he-lix_toolbox with a helical rise search range of ±10 grid points of 0.088 Å and a helical twist search range of ±20 grid points of 0.014° (note that because we assumed D14 symmetry a priori, we only searched for one set of helical parameters). (iv) Symmetrization of the maps by first

applying D14 symmetry using relion_image_handler followed by applying helical symmetry using relion_helix_toolbox with the refined helical parameters (--cyl_inner_diameter 380 --cyl_outer_diameter 501 --z_percentage 0.3 –sphere_percentage 0.9 –width 5). (v) Calculation of FSC curves from the half maps. For this, we split the segments equally in the middle of the particle stack, in order to avoid contribution of segments extracted from the same VP39 tube to both half maps. (vi) We performed eight iterations of steps (i) to (v). (vi) Refinement of CTF parameters with relion_ctf_refine, where we used per-particle defocus fitting; beam-tilt refinement, and fitting of anisotropic magnification distortion for each of the particle stack's 18 optics groups. (vii) Refinement of particle images with relion_motion_refine. After two iterations of this protocol, we observed convergence of the calculated FSC curves (Supplementary Fig. 4A, C). The helical rise refined to 43.86 Å and the helical twist to −7.16° (Supplementary Table 1).

### Local alignment and reconstruction of VP39 dimers

For local reconstruction, we symmetry expanded the 4983 segment particle stack metadata file with relion_particle_symmetry_expand in two steps. First, with --helix --twist 43.86 --rise -7.16 --asu 1. Second, with --sym C14. This yielded 42 particles per segment, or an expanded stack of 209,286 images. After signal subtraction with relion_project of the entire helical density, except a single set of four adjacent VP39 dimers, we obtained 209,286 signal-subtracted segment images. Since sub-particle extraction from the motion-refine corrected stack was not possible in Relion, we extracted subparticles with a box size of 360 pixels from the 209,286 signal-subtraced segment images using cus-tom made Python scripts and IMOD programs[73]. The subparticle box was centered on the center of gravity of the four VP39 dimers. We used alignment by classification[52], followed by local alignment with cisTEM[69] and local density averaging of the eight protomers with relion_lo-calsym. We used sharpen_map from cisTEM for postprocessing and map sharpening. The FSC for the local reconstruction at different processing steps is shown in Supplementary Fig. 4B, D. The final resolution was 3.2 Å (Supplementary Fig. 4D) as judged by correlation between the half maps (Supplementary Table 1).

### Supervised classification of helical segments

In order to determine the helical symmetry and extent of potential tube flattening upon sample vitrification for all our extracted seg-ments, we prepared a set of 3D references with defined helical geo-metry and flattening for supervised classification. We note that caution should be taken when applying this supervised classification approach, as it may generate incorrect symmetry assignments along with assignments that are correct. The results of our supervised classifica-tion require careful analysis.

We first calculated the unit cell vectors $\mathbf{a}$ = [27.91 Å, −43.86 Å] and $\mathbf{b}$ = [72.31 Å, 43.86 Å] of the underlying 2D crystal lattice that described the [$n_1$ = 14, $n_2$ = 14]-symmetry of our 4.1 Å-resolution reconstruction. Different helical symmetries, where the local packing of VP39 proto-mers is invariant (or almost invariant) by obeying the same 2D crystal lattice and where the resulting tubes do not contain a seam, can be defined by a wrapping vector $\mathbf{w} = n_1\mathbf{a} + n_2\mathbf{b}$[74]. The wrapping vector defines how the 2D crystal lattice is wrapped into a tube. It lies in the equator of the helix and its length corresponds to the circumference of the helix (Fig. 5A). Thus, with $\mathbf{a}$ and $\mathbf{b}$ defined, we calculated the helical parameters $x_1, y_1, x_2, y_2$ for any chosen lattice point [$n_1, n_2$] by vector analysis and trigonometry using NumPy[72] (Fig. 5B, C). $x_1$ and $x_2$ are the arc lengths with corresponding helical twist$_1$ and twist$_2$, respectively. $y_1$ and $y_2$ are the helical rise$_1$ and rise$_2$, respectively. If $n_1$ and $n_2$ are integer times of each other, the helical packing results in rotational symmetry ($x_1 = x_2$ and $y_1 = y_2$), e.g., C14 for the [$n_1$ = 14, $n_2$ = 14]-symmetry (Fig. 5B). If $n_1$ and $n_2$ are different, there is no rotational symmetry (Fig. 5C).

We rigid-body fitted the refined VP39 dimer structure into the map of the [$n_1$ = 14, $n_2$ = 14]-symmetry reconstruction and then used

PyMOL to symmetry-expand it according to the helical geometries for a chosen set of wrapping vectors in order to generate atomic models of the full tubes (Fig. 5D). The lengths of the wrapping vectors for which we calculated 3D references were between 1140 and 1660 Å. This translated to diameters between approximately 360 and 530 Å, reflecting the observed distribution in diameters of our imaged VP39 tubes (Supplementary Fig. 2). We first radially translated the dimer to match its center of mass to the radius of the helix. Symmetry expansion then involved rotation and translation according to the calculated helical twists and rises, followed by a rotation around the radial vector of the dimer to account for the relative orientation of the 2D lattice for any given wrapping vector (Fig. 5B, C). To model flattening of the tubes, we introduced deformation to elliptical helical geometry as previously described[52]. We used a scale of 1.00–1.09 for the major axis of the ellipse and made ten 3D references for each wrapping vector (0–9% flattening) (Fig. 5B, C). We used SciPy[75] for calculation of elliptical line integrals and to numerically determine the minor axis for a given elliptical circumference. We matched helical twists to elliptical path lengths (putting the dimer center of gravity to the correct x and y coordinates) and kept a reference vector normal to the tangential plane of the elliptical helix (rotation of the dimer around an axis along z and going through its center of gravity). An animation of the deformation of the $[n_1 = 14, n_2 = 14]$-symmetry tube is shown in Supplementary Movie 1. Supplementary Movie 2 shows the power spectra of the least and the most flattened classes for segments corresponding to the $[n_1 = 14, n_2 = 14]$-symmetry. We used the programs sfall and fftbig from CCP4[76] to calculate structure factors and density maps from the modeled tubes (box size = 912). This yielded 1440 3D references for supervised classification.

We used a supercluster computer where we had access to about 4000 CPUs for parallelized classification of the full segment particle stack with 72,013 images. With refine3d from cisTEM[69], we globally aligned each segment to each of the 1440 3D references using two-times binned data. We limited the resolution for alignment to 12 Å (dictated by the available computational resources), imposed C1 symmetry, and used a spherical mask with an outer radius of 360 Å. We assigned each segment to the 3D reference that gave the highest score and mapped the distribution onto the 2D crystal lattice shown in Fig. 5D. To verify the helical symmetry assignments of segments after supervised classification, we compared the direct fit between the averaged power spectra of the segments and the averaged power spectra calculated from projections of the corresponding 3D references (Supplementary Movies 3 and 4, Supplementary Fig. 14). Supplementary Fig. 15 shows the class partitioning of subparticles derived from either 0% or 2% flattened segments after classification without alignment. Given the challenges and ambiguities associated with Fourier-Bessel indexing[56,57], we also calculated a reconstruction using all segments that partitioned into the next most populated 0–2% flattened classes of $[n_1 = 13, n_2 = 14]$-symmetry (Supplementary Fig. 16). For this, we used a simplified reconstruction protocol, in which we followed the main protocol for refined helical reconstruction but only performed one iteration (instead of two) of alignment with cisTEM, reconstruction with C1 symmetry, map symmetrization, and refinement of CTF parameters with relion_ctf_refine. In contrast to the main reconstruction protocol, where we symmetrized the maps by first applying D14 symmetry, we used local symmetry instead, followed by applying helical symmetry as done in the main reconstruction protocol. To compare the simplified reconstruction protocol with the main reconstruction protocol, we applied this simplified protocol to all 19,012 segments of the main $[n_1 = 14, n_2 = 14]$-symmetric reconstruction and obtained a reconstruction at 4.1 Å (3.6 Å with the main reconstruction protocol). To compare the effect of having only a limited amount of segments available, as is the case for the $[n_1 = 13, n_2 = 14]$-class (1349 segments), we randomly selected 1,196 segments from the $[n_1 = 14, n_2 = 14]$-symmetric class and calculated a

reconstruction using the simplified protocol, which yielded a resolution of 9.2 Å, showing that limiting the number of segments to about 1200 does not even yield a high-resolution reconstruction for the $[n_1 = 14, n_2 = 14]$-class.

To improve the helical and local reconstructions, we used the supervised classification approach to increase the number of segments for reconstruction by including segments that corresponded to 1% and 2% flattened tubes in addition to the non-flattened tubes. 19,012 segments, which classified with the 3D reference of $[n_1 = 14, n_2 = 14]$-symmetry and corresponded to 0–2% tube flattening, were used for a final round of helical reconstruction, followed by local reconstruction (Supplementary Figs. 1 and 4 for FSC curves).

## Model building and refinement

Automated model building was performed by ModelAngelo[44], using the locally-averaged electron density map in the absence of a sequence input. Following manual model building in Coot[45], we refined the model using real-space refinement[77,78] and validated it using Mol-Probity in Phenix[79]. Protein structure comparison was performed using the Dali Protein Structure Comparison Server[46] and the Foldseek Search Server[47]. Molecular representations were generated and analyzed with PyMOL (The PyMOL Molecular Graphics System, Version 2.5 Schrödinger, LLC) and UCSF ChimeraX (Resource for Biocomputing, Visualization, and Informatics, University of California, San Francisco)[80].

## Sequence alignment and conservation mapping

We collected 81 amino acid sequences of 61 alphabaculoviruses, 17 betabaculoviruses, 2 gammabaculoviruses, and 1 deltabaculovirus using NCBI BLAST[81,82]. Sequences were aligned using MAFFT (online webserver)[83,84] and alignments were analyzed and edited in Jalview v. 2.11.2.6[85]. Sequences with long gaps or insertions were removed prior to performing a second round of MAFFT alignment. The final alignment included 73 sequences. Sequence conservation was mapped onto the VP39 structure using the ConSurf webserver[86,87]. Alignments were visualized with EMBL's multiple sequence alignment viewer MView[88]. Phylogenetic trees were calculated in Jalview using the Neighbor Joining algorithm with a BLOSUM62 substitution matrix[89] and visualized using the interactive Tree of Life iTOL[90].

## Structure prediction

Model predictions for baculoviral VP39 dimers were calculated by AlphaFold2-multimer[50,51] using ColabFold v. 1.5.2[91] and the COSMIC[2] platform[92]. Predicted structures were compared to the reconstructed VP39 dimer by superposition onto the monomer in the dimer structures using the super command in PyMOL.

## Mass spectrometry

Purified nucleocapsids in 20 mM HEPES, 150 mM NaCl, 0.02% (w/v) LMNG, pH 7.5 were denatured in 5% (w/v) SDS and reduced with 0.2 M DTT at 57 °C for one hour. Following alkylation with 0.5 M iodoaceta-mide for 45 min in the dark, samples were acidified with 12% phosphoric acid for a final concentration of 1.2% phosphoric acid. Upon loading of seven-fold diluted sample in S-trap binding buffer (90% methanol/100 mM TEAB) to S-trap spin column (Protifi, NY, USA), the sample was washed three times and digested with 500 ng trypsin at 47 °C for one hour. Peptides were eluted by the addition of 40% (v/v) acetonitrile in 0.5% (v/v) acetic acid, followed by the addition of 80% (v/v) acetonitrile in 0.5% (v/v) acetic acid. The organic solvent was removed using a SpeedVac concentrator and the sample was reconstituted in 0.5 (v/v) acetic acid.

Approximately 1 µg of each sample was analyzed individually by LC/MS/MS. Samples were separated online using a Thermo Fisher Scientific EASY-nLC 1200 where solvent A was 2% acetonitrile/0.5% acetic acid and solvent B was 80% acetonitrile/0.5% acetic acid. A

120 min gradient from 5 to 35% B was applied for all samples. Peptides were gradient eluted directly to a Thermo Fisher Scientific Orbitrap Eclipse Mass Spectrometer. High resolution full spectra were acquired with a resolution of 120,000, an AGC target of 4e5, with a maximum ion time of 50 ms, and a scan range of 400–1500 m/z. All precursors with charge states between 2 and 10 were selected for fragmentation. Dynamic exclusion was set for 30 s after one scan. All MS/MS of spectra were collected with a resolution of 30,000, AGC target of 2e5, maximum ion time of 200 ms, one microscan, 2 m/z isolation window, auto scan range mode, and NCE of 27. The instrument was set to run at top speed with a cycle time of 3 s. Samples were run a second time with the same gradient and MS parameters except only charge states between 4 and 10 were selected for fragmentation. MS/MS spectra were searched in PD1.4 against the *Autographa*, *Bombyx*, *Spodoptera frugiperda*, and contaminants databases. MS/MS spectra were also searched in Byos by Protein Metrics against the Major Capsid Protein and VP39 sequences alone to find peptides.

### Reporting summary

Further information on research design is available in the Nature Portfolio Reporting Summary linked to this article.

## Data availability

The cryo-EM maps generated in this study have been deposited in the Electron Microscopy Data Bank under accession code EMD-41133. The refined model coordinates generated in this study have been deposited in the Protein Data Bank under accession codes 8TAF. VP39 Protein sequences used in this study are available in the UniProt database and the NCBI reference sequence database RefSeq and their accession codes are provided in Supplementary Table 6. All data are available from the corresponding author upon request. Source data are provided as a Source Data file. Source data are provided with this paper.

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

## Acknowledgements

We thank Z. Li, S. Sterling, R. Walsh, M. Mayer, R. Nair, and S. Rawson from the Harvard Medical School Cryo-EM Center for Structural Biology and the Harvard Medical School Molecular Electron Microscopy Suite for their exceptional support during grid screening and data collection; M. Rigney and B. Isaac for greatly facilitating our use of the Brandeis Electron Microscopy Facility; the SBGrid Consortium for assistance with software and high-performance computing; J. Delgado from the MGH Department of Molecular Biology for support with software and high-performance computing; M. Antonelli and B. Ueberheide from the Proteomics Center at New York University Grossman School of Medicine for mass spectrometry analysis of our samples; T. Grant for discussions and advice; G. Lander, M. Ohi, J. Kollman, M. Vos, and the Helmsley Charitable Trust for the opportunity to attend the 2019 Cryo-Electron Microscopy course at Cold Spring Harbor Laboratories (F.M.C.B.); and Y. Han, X. Fan, E. Eng, A. Noble and the staff from the New York Structural Biology Center for the opportunity to attend the workshop on graphene grids for CryoEM hosted by the National Center for Cryo-EM Access and Training and the National Resource for Automated Molecular Microscopy (F.M.C.B.). Portions of this research were conducted on the O2 High Performance Compute Cluster, operated by the Research Computing Group at Harvard Medical School. We acknowledge support from the Swiss National Science Foundation grant P180777 (F.M.C.B.), and the National Institutes of Health R35GM130289 (X.Z.) and R35GM142553 (L.H.C.)

## Author contributions

F.M.C.B. and L.H.C. conceived of the study and designed the experiments. F.M.C.B. and T.H.N. expressed the proteins. F.M.C.B. purified the nucleocapsids. F.M.C.B. collected the data and together with C.Y.G. and L.H.C. picked the segments. F.M.C.B., C.Y.G., S.J., X.Z. and L.H.C. analyzed the data, reconstructed the structures, and built models. F.M.C.B. and L.H.C. wrote the manuscript with contributions from S.J. All authors approved the final version of the manuscript.

## Competing interests

The authors declare no competing interests.
