## [Peer Review File · Nature Communications]

Helical reconstruction of VP39 reveals principles for baculovirus nucleocapsid assemblyREVIEWER COMMENTS

Reviewer #1 (Remarks to the Author):

This is a potentially interesting structural study of the VP39 nucleocapsid. There are a number of minor issues, but the most major one is that there is no discussion of any relation between this structure and the viral genome. It is called a nucleocapsid for a reason. The abstract states that “Analysis of sample polymorphism revealed tube flattening could account for different helical geometries.”

But the paper states the exact opposite, such as: “Most of the remaining segments belonged to tubes of either C15 or C13 symmetry, which correspond to occasional insertions or deletions of single subunits in the helical plane. These differences in helical symmetries generate nucleocapsids that match the published range of AcMNPV diameters.”

and

“which correspond to our observations of 13- to 15-start helices by indexing of segment 2D class averages (Extended Data Fig. 1, Supplementary Fig. 2).” Perhaps the abstract and text were written by two different people? Or the abstract was written before the analysis was done?

More minor issues:

Lines 128-129) It is stated that the resolution of 4.0 Å was obtained by a map:model FSC. But in Methods, it is stated: “resulted in a map at about 4 Å resolution (3.3 Å FSC at 0.148 in CryoSPARC using half maps, 4.3 Å FSC at 0.5 map vs model”. So what is stated in the text appears not to be true. Further, it is worth noting that the so-called “gold standard” map:map FSC is not really a gold standard, as shown by this example. It is stated in the legend to Extended Data Fig. 2 that the curves “do not strictly drop to zero” but this is misleading, as they simply do not drop to zero showing the artifactual correlation between the two half maps.

Line 133) “spiral shape” is a poor description, as a spiral typically has a changing radius. It would be much better to simply refer to these as helical strands.

Line 155) “each monomer”. There are no monomers, which by definition are monomeric. This should be “each subunit”

Lines 396-397) “majority of the tubes (40% of the segments”. A majority means > 50%!

Lines 401-403) “Our analysis furthermore indicated that most of the tubes were flattened in our preparation, which may be due to blotting during sample vitrification and the flexibility of the inter-dimer contacts.” There is no explanation here how blotting could cause such flattening. What is more likely is that the large compressional forces arising from the thin film in which the tubes are embedded are responsible for the flattening. This has been described previously for axonemes (e.g., Bui et al., 2012, JCB 198: 913-925).

Line 655) “We limited to resolution for alignment” should be “We limited the resolution for alignment”.

Reviewer #2 (Remarks to the Author):

In their manuscript “Helical reconstruction of VP39 baculovirus nucleocapsid assembly reveals principles for baculovirus nucleocapsid assembly”, the authors report the structure of the filament forming main component of the nucleocapsid, VP39, from *Autographa californica* multiple nucleopolyhedrosis virus. While the thorough analysis of the structural data and the link to known biological data are clear strengths of this manuscript, several image processing methodological points need to be addressed to validate all conclusions drawn by the authors.

Below a detailed review of the points of concern.

Major remarks:

A general comment is that the processing strategy chosen by the authors is unnecessarily complicated, and the interplay between different processing software (cisTEM, Cryosparc, Relion..) can be a source of problems, such as loss of useful data (such as the correspondence between filament IDs and segments, intersegment distances, etc.), or the observed artifacts in the FSC curve (not dropping to zero). Based on the data description and the difficulties of this dataset, there is no clear justification (which could be added by the authors in the Method section) for choosing this processing strategy, rather than doing the entire processing in Relion (or Cryosparc), that include all tools for helical refinement.

A.1/ In the “Data availability” section, the authors mention that both the helical map and the local refinement map were deposited. However, in the PDB validation report, we can only see the local refinement map, this should be fixed and a new report sent for review.

Based on other abnormalities in the validation report (see below in the minor points), we kindly ask the authors to provide the half maps for further review.

A.2/ None of the half maps FSCs are dropping to zero. The authors justify that by “because of the semi-independent half map refinement as implemented in in cisTEM, the high degree of symmetry, and the relatively small masked volume compared to the total volume of the box, there likely is some inflation of the curves”. Based on the curve shapes, the inflation is rather “certainly” than “likely”, and the justifications proposed by authors do not seem appropriate, and should be re-phrased, if not fixed (which is the preferred recommendation, but the reviewer appreciate the amount of work for getting a final structure that will look very similar to the presented one).

First, for the curves corresponding to helical reconstruction in Extended Data Fig 2A, FSCs should be calculated between unsymmetrized half maps, as it is done in Relion or Cryosparc, not between symmetrized half maps. A mask containing only a small z portion of the helix can be used to avoid including blurry edges of the unsymmetrized half maps in the FSC calculation.

For the helical reconstruction again, when converting from cisTEM to Cryosparc or to Relion, if the correspondence between filaments IDs and segments is lost, or if the information on the distance between the segments from extraction is lost, it is expected to get such strong artifacts in the FSCs. To avoid that, re-extraction should be done so that the respective software has all the necessary metadata on the segments, used to split the dataset appropriately, and to limit the Y search of each segment based on the intersegment distance, which are the likely reasons of the FSC curves not dropping.

For the local refinements (Extended Data Fig 2B), the authors used the symmetry expansion tool `relion_particle_symmetry_expand` with the option `--asu 3` (as stated in the method). Based on the extraction parameters, the “asu” option should have been 1, otherwise there is data duplication which can be an additional reason for the artifactual FSC curves (on top of the reasons mentioned for the helical refinement).

The justifications on “high degree of symmetry, and the relatively small masked volume” should be removed, since those properties are present for many helical assemblies for which such artifacts are not present.

A.3/ The analysis of the nucleocapsid polymorphism by a supervised classification approach, although interesting and well thought -especially the reference preparation part-, suffer from severe drawbacks that make it difficult to take the author’s conclusions as granted.

First the authors should provide evidence of the correctness of the different helical symmetry assignments : this shouldn’t be difficult, since the classification provide them with different segment stacks and associated helical symmetry that could be used for refinement to provide high -or medium- resolution maps corresponding to the the different helical symmetries. On top of validating the classification procedure, this would allow the author to merge subtracted particles from different diameter tubes, hence improving the resolution of the local refinements and better characterizing potential lattice deformation (stretching, etc) by 3D classification or 3D variability analysis of the subtracted particles.

Another simple analysis that would strengthen the approach, would be to show the orientation distribution (on-axis and out-of-plane angles), for some representative references (e.g. the 5 most populated without flattening). The distribution is expected to be ~flat for on-axis angles, and gaussian-shaped, centered on 0 (or 90 depending on the convention) for the out-of-plane angles, if the symmetry assignment is correct. If the first proposed approach (calculating the 3D structures corresponding to the different helical symmetries) is beyond the scope of this work on author’s point of view, then the authors should provide these plots for validation.

For the flattening, to be more conclusive, the authors could show the sum of power spectra of aligned segments corresponding to a particular helical symmetry (e.g. the most abundant), for the least and the most flattened classes. Effects of the flattening should be visible by the stretching of the signal on the layer lines.

Lines 345-347, the authors state that “A closer inspection of the segments belonging to tubes with [n1=14, n2=14]-symmetry revealed that only approximately 14% of these segments aligned best to the non-flattened 3D reference.” Have the authors tried to run refinements before and after removing flattened segments, e.g. keeping up to flattening of 5%, and assess the improvements ? On one hand, this would validate the classification based on flattening, and on the other hand integrate this classification procedure in their processing pipeline.

A.4/ Lines 572-587 : the authors describe a complicated protocol for refined helical reconstruction, instead of simply using `relion_refine` with the helical options . Could the authors explain why ? If the

reason is the loss of metadata (priors on in-plane and out-of-plane, intersegment distance, filament IDs), then the authors should consider re-extracting in Relion using the filament ends coordinates corresponding to their subset of selected segments.

Minor points :

B.1/ Lines 144-149 : “We initially used ModelAngelo [...] in Phenix”. Most of this paragraph seem more appropriate for the method section.

B.2/ Lines 132-133 : “Our reconstruction reveals that individual VP39 subunits pack as dimers that assemble into 14 protofilaments”. This is a general statement that does not apply to the entire dataset, it should be more precisely phrased for the readers to understand that it also assembles in other number of protofilaments structures.

B.3/ Line 532 : “with a search range of -15° to 15° , rise: 43.8 \AA with a search range of $20\text{-}60 \text{ \AA}$, C15”. This should be C14.

B.4/ Supplementary Fig 2A : “Power spectrum of a 2D class average, which has been symmetrized along its meridian”. Instead of the PS of the 2D class average, it is recommended to use the sum of PS of the segments belonging to that class-average, which avoid some artifacts present in the PS of class-averages, such as left/right asymmetry. No symmetrization should be necessary if the PS is not artifactual.

B.5/ Lines 315-316 : “potential symmetries (C12-C15) possible for the given Bessel orders identified during Fourier-Bessel analysis of several 2D class averages (Supplementary Fig. 2).” Supp Fig 2 shows only the analysis on one class, therefore the reader can not appreciate the validity of this statement and the differences in first Bessel peak position across the different structures (which might actually be compensated by diameter variation).

B.6/ Lines 544-545 : “We determined initial helical symmetry parameters by indexing the power spectra of these class averages using PyHI (Python v. 3.7)”. It is not clear why this indexation was done, since the authors already obtained helical parameters in previous steps.

B.7/ For the local refinements, it should be stated whether refinement was done with C2 (or D1) symmetry, or only in C1. Since the symmetry expansion was done with C14 and not D14, we would recommend the authors to use the 2-fold symmetry for refinement.

B.8/ Validation report, section 6.2.1/6.2.2 : While the “raw map” seem to have 2-fold cyclic symmetry, or at least have one of its axis (X) aligned with a (maybe imposed, see point B.7 to clarify) C2 symmetry axis, this is not the case for the “primary map”. Why such a difference ?

In the “raw map” central slices, can the author explain the very strong differences in areas with no protein density near the center and the areas with no protein density far from the center ? In other words, why is the noise appearance so different depending on the area of the volume ? It looks like a different low-pass filter had been applied in different areas, which seem odd for a “raw map”.

Reviewer #1 (Remarks to the Author):

This is a potentially interesting structural study of the VP39 nucleocapsid. There are a number of minor issues, but the most major one is that there is no discussion of any relation between this structure and the viral genome. It is called a nucleocapsid for a reason.

We appreciate the reviewer's comment regarding discussing the relationship between the structure and viral genome packaging. We would like to point out that we write in the Discussion section: "The fold includes a Zn-finger (ZF) region with a Zn²⁺ ion coordinated by a conserved CCCH motif, which is facing the capsid lumen. Moreover, it is surrounded by basic residues, **priming this region as a possible binding pocket for the viral DNA.**" In the Results section, we describe that "The ZF faces the luminal volume of the VP39 nucleocapsid assembly and is surrounded by positively charged residues, **consistent with a role in binding viral DNA** (Fig. 2E; Extended Data Fig. 5, Extended Data Fig. 6)." While we can speculate as to the location of a potential DNA binding site, we are cautious in making any more claims without extensive experimental evidence.

We have now added the following sentence to the discussion: "**The charge pattern across the luminal face of the nucleocapsid suggests a surface primed for interacting with multiple strands of packaged DNA (Extended Data Fig. 6).**"

The question of viral genome packaging into the nucleocapsid, while a very interesting question, is beyond the scope of our work. In our dataset and in unpublished negative stain micrographs, all tubes appeared to be empty. Investigating the interaction of the viral genome and the structure would require obtaining either fully packaged nucleocapsids before genome release, or nucleocapsids, which are in the process of being packaged with DNA. This requires catching the nucleocapsid in just the right time in its complex life cycle. We hope that the reviewer can appreciate that our structure—and especially our discovery of an inward-facing Zn-binding site in VP39—now opens new experimental avenues for investigating DNA binding and packaging into the nucleocapsid.

The abstract states that "Analysis of sample polymorphism revealed tube flattening could account for different helical geometries." But the paper states the exact opposite, such as: "Most of the remaining segments belonged to tubes of either C15 or C13 symmetry, which correspond to occasional insertions or deletions of single subunits in the helical plane. These differences in helical symmetries generate nucleocapsids that match the published range of AcMNPV diameters."

and

"which correspond to our observations of 13- to 15-start helices by indexing of segment 2D class averages (Extended Data Fig. 1, Supplementary Fig. 2)." Perhaps the abstract and text were written by two different people? Or the abstract was written before the analysis was done?

We thank the reviewer for pointing out that statements in the abstract and the main text regarding sample polymorphism appear contradictory. We updated the sentence in the abstract to "Analysis of sample polymorphism revealed that VP39 **assembles in several closely-related helical geometries.**

More minor issues: Lines 128-129) It is stated that the resolution of 4.0 Å was obtained by a map:model FSC. But in Methods, it is stated: "resulted in a map at about 4 Å resolution (3.3 Å

FSC at 0.148 in CryoSPARC using half maps, 4.3 Å FSC at 0.5 map vs model". So what is stated in the text appears not to be true. Further, it is worth noting that the so-called "gold standard" map:map FSC is not really a gold standard, as shown by this example. It is stated in the legend to Extended Data Fig. 2 that the curves "do not strictly drop to zero" but this is misleading, as they simply do not drop to zero showing the artifactual correlation between the two half maps.

As suggested by reviewer #2, we have now calculated new reconstructions from a segment particle stack with [$n_1=14$, $n_2=14$] symmetry after supervised classification (see below). We have also fixed an error in our code base that caused inflated correlation in the FSC curves calculated from half maps (there was a fraction of tubes that contributed some segments to both half maps); they do drop to zero now as would be expected. Reviewer #1 is correct that the half map FSC curves are not "gold standard", because we used cisTEM for particle alignment, where the half maps are treated semi-independently, but where the resolution is limited to prevent alignment bias.

We have updated Extended Data Fig. 2 with the corrected FSC curves from the new reconstructions and refined model. The estimated nominal resolution from the half map analysis (3.6 Å for the helical reconstruction, 3.2 Å for the local reconstruction) is now in much better agreement with the values obtained from the full map to model analysis (4.0 Å for the helical reconstruction, 3.3 Å for the local reconstruction).

We updated the text in Methods to reflect the correct value of 4.0 Å FSC at 0.5 map vs model, as shown in Extended Data Fig. 2C: "Following one round of 3D classification (10 classes), in which we selected two classes with similar tube diameter and refined each class separately using the helical refinement option in CryoSPARC (twist: 7.5° with a search range of -15° to 15°, rise: 44 Å with a search range of 20-60 Å, D14, non-uniform refinement option), a final helical refinement of the two combined classes (4,983 segments, twist: 7.5° with a search range of -15° to 15°, rise: 44 Å with a search range of 20-60 Å, D14, non-uniform refinement option), resulted in a map at 4.1 Å resolution (0.143 FSC at 4.1 Å using half maps, 0.5 FSC at 4.4 Å using map and final model [see below]; Extended Data Fig. 2A and C)."

Line 133) "spiral shape" is a poor description, as a spiral typically has a changing radius. It would be much better to simply refer to these as helical strands.

We updated the text in line 133 to "In our reconstruction, individual VP39 subunits pack as dimers that assemble into 14 helical strands and together form the central cylindrical structure of the baculoviral nucleocapsid (Fig. 1C)."

Line 155) "each monomer". There are no monomers, which by definition are monomeric. This should be "each subunit"

We updated the text in line 155 to "Each dimer subunit comprises a mixed alpha/beta fold, with extensive interdigitation of elements with its partner in the dimer (Fig. 2B, D)." Further, in the discussion, we changed the text to "The VP39 dimer subunit adopts a unique mixed alpha/beta fold (Fig. 2), which was not recognized by the DALI structural comparison server and Foldseek."

Lines 396-397) "majority of the tubes (40% of the segments)". A majority means > 50%!

We updated the text in lines 396-397 to "We found that 40% of the tubes in our dataset assembled with [$n_1=14$, $n_2=14$] helical symmetry."

Lines 401-403) “Our analysis furthermore indicated that most of the tubes were flattened in our preparation, which may be due to blotting during sample vitrification and the flexibility of the inter-dimer contacts.” There is no explanation here how blotting could cause such flattening. What is more likely is that the large compressional forces arising from the thin film in which the tubes are embedded are responsible for the flattening. This has been described previously for axonemes (e.g., Bui et al., 2012, JCB 198: 913-925).

We thank the reviewer for raising this point. We agree that there is no explanation for how blotting could cause such flattening and we appreciate the provided reference for axonemes. We changed the sentence in lines 401-403 to: “Our analysis furthermore indicated that most of the tubes were flattened in our preparation. **While we cannot unambiguously identify the source of flattening in our sample, it is noteworthy that tube flattening has been observed for cilia in published work⁶⁰, with one study describing tube compression parallel to the ice plane on the EM grid⁶¹.**

Ref 60: Bui *et al.*, 2012, JCB 198: 913-925

Ref 61: Greenan *et al.*, 2020, JCB 219:e201907060

Line 655) “We limited to resolution for alignment” should be “We limited the resolution for alignment”.

We updated the text in line 655 to “We limited the resolution for alignment to 12 Å (dictated by the available computational resources), imposed C1 symmetry, and used a spherical mask with an outer radius of 360 Å.”

Reviewer #2 (Remarks to the Author):

In their manuscript “Helical reconstruction of VP39 baculovirus nucleocapsid assembly reveals principles for baculovirus nucleocapsid assembly”, the authors report the structure of the filament forming main component of the nucleocapsid, VP39, from *Autographa californica* multiple nucleopolyhedrosis virus. While the thorough analysis of the structural data and the link to known biological data are clear strengths of this manuscript, **several image processing methodological points** need to be addressed to validate all conclusions drawn by the authors. Below a detailed review of the points of concern.

Major remarks:

A general comment is that the processing strategy chosen by the authors is unnecessarily complicated, and the interplay between different processing software (cisTEM, Cryosparc, Relion..) can be a source of problems, such as loss of useful data (such as the correspondence between filament IDs and segments, intersegment distances, etc..), or the observed artifacts in the FSC curve (not dropping to zero). Based on the data description and the difficulties of this dataset, there is no clear justification (which could be added by the authors in the Method section) for choosing this processing strategy, rather than doing the entire processing in Relion (or Cryosparc), that include all tools for helical refinement.

On the processing strategy:

The complexity of this structure determination required the use of program tools from different software packages. We made sure that all metadata was carried forward when switching from one package to the other (including all the metadata associated with helical reconstructions, such as the filament ID, helical track length, and prior angles). We provide a detailed explanation in A.4 and we have further clarified the chosen processing strategy in the Methods section.

On the FSC curves:

We were able to fix an error in our code base that caused the observed artifacts in the FSC curves, and they drop to zero now as expected (see comment in A.2 below).

A.1/ In the “Data availability” section, the authors mention that both the helical map and the local refinement map were deposited. However, in the PDB validation report, we can only see the local refinement map, this should be fixed and a new report sent for review. Based on other abnormalities in the validation report (see below in the minor points), we kindly ask the authors to provide the half maps for further review.

As suggested below, we have now calculated new reconstructions from a segment particle stack with $[n_1=14, n_2=14]$ symmetry after supervised classification (see below) and have deposited all relevant maps (helical and local reconstructions) together with the refined model. We are also happy to make all these files available for further review. They are the following:

Helical reconstruction:

Unmodified maps (reconstructed in C1):

- NCOMMS-23-27865-T_aug15_helical.mrc
- NCOMMS-23-27865-T_aug15_helical_half_map1.mrc
- NCOMMS-23-27865-T_aug15_helical_half_map2.mrc

Symmetrized maps (helical symmetry applied):

- NCOMMS-23-27865-T_aug15_helical_sym.mrc
- NCOMMS-23-27865-T_aug15_helical_half_map1_sym.mrc
- NCOMMS-23-27865-T_aug15_helical_half_map2_sym.mrc

Mask (used for FSC calculation):

- NCOMMS-23-27865-T_aug15_helical_mask_fsc.mrc

Symmetrized and masked maps:

- NCOMMS-23-27865-T_aug15_helical_sym_masked.mrc
- NCOMMS-23-27865-T_aug15_helical_half_map1_sym_masked.mrc
- NCOMMS-23-27865-T_aug15_helical_half_map2_sym_masked.mrc

Mask (used for sharpening):

- NCOMMS-23-27865-T_aug15_helical_mask_sharp.mrc

Symmetrized, masked and sharpened map:

- NCOMMS-23-27865-T_aug15_helical_sym_masked_sharp_3.6.mrc

Local reconstruction:

Unmodified maps (reconstructed in C1):

- NCOMMS-23-27865-T_aug15_local.mrc
- NCOMMS-23-27865-T_aug15_local_half_map1.mrc
- NCOMMS-23-27865-T_aug15_local_half_map2.mrc

Symmetrized maps (local symmetry applied):

- NCOMMS-23-27865-T_aug15_local_sym.mrc
- NCOMMS-23-27865-T_aug15_local_half_map1_sym.mrc
- NCOMMS-23-27865-T_aug15_local_half_map2_sym.mrc

Mask (used for FSC calculation):

- NCOMMS-23-27865-T_aug15_local_mask_fsc.mrc

Symmetrized and masked maps:

- NCOMMS-23-27865-T_aug15_local_sym_masked.mrc
- NCOMMS-23-27865-T_aug15_local_half_map1_sym_masked.mrc
- NCOMMS-23-27865-T_aug15_local_half_map2_sym_masked.mrc

Mask (used for sharpening):

- NCOMMS-23-27865-T_aug15_local_mask_sharp.mrc

Symmetrized, masked and sharpened map:

- NCOMMS-23-27865-T_aug15_helical_sym_masked_sharp_3.2.mrc
- NCOMMS-23-27865-T_aug15_helical_sym_masked_sharp_3.0.mrc
- NCOMMS-23-27865-T_aug15_helical_sym_masked_sharp_2.8.mrc
- NCOMMS-23-27865-T_aug15_helical_sym_masked_sharp_2.6.mrc

Refined model:

- real_space_refined_001.pdb

A.2/ None of the half maps FSCs are dropping to zero. The authors justify that by “because of the semi-independent half map refinement as implemented in in cisTEM, the high degree of symmetry, and the relatively small masked volume compared to the total volume of the box, there likely is some inflation of the curves”. Based on the curve shapes, the inflation is rather “certainly” than “likely”, and the justifications proposed by authors do not seem appropriate, and should be re-phrased, if not fixed (which is the preferred recommendation, but the reviewer appreciate the amount of work for getting a final structure that will look very similar to the presented one). First, for the curves corresponding to helical reconstruction in Extended Data Fig 2A, FSCs should be calculated between unsymmetrized half maps, as it is done in Relion or Cryosparc, not between symmetrized half maps. A mask containing only a small z portion of the helix can be used to avoid including blurry edges of the unsymmetrized half maps in the FSC calculation.

We are grateful that the reviewer pointed this out. We investigated and discovered an error in our code base that we have fixed now. Specifically, there was a fraction of tubes that contributed segments to both half maps, which lead to the observed inflation of the FSC curves. We have recalculated the FSC curves with the updated code and they do now drop to zero as can be seen in the revised Extended Data Fig. 2. The estimated resolution based on the half maps (3.6 Å for the helical reconstruction, 3.2 Å for the local reconstruction) is now also in agreement with the estimated resolution based on the correlation between the refined models and the final maps (4.0 Å for the helical reconstruction, 3.3 Å for the local reconstruction). Because we used cisTEM for particle alignment, where half maps are treated “semi-independently” and alignment bias is prevented by limiting the resolution for alignment, our analysis was not affected by this code error except for the final FSC calculation. We have removed the wrong explanation for the inflated FSC curves, which is now obsolete, from the manuscript.

To our knowledge, Relion and Cryosparc calculate FSC curves after symmetrization, we therefore believe it would not be helpful to add such plots to the manuscript. However, we have now deposited (and made available to the reviewer) the unmodified, unsymmetrized half maps that were reconstructed in C1.

For the helical reconstruction again, when converting from cisTEM to Cryosparc or to Relion, if the correspondence between filaments IDs and segments is lost, or if the information on the distance between the segments from extraction is lost, it is expected to get such strong artifacts in the FSCs. To avoid that, re-extraction should be done so that the respective software has all the necessary metadata on the segments, used to split the dataset appropriately, and to limit the Y search of each segment based on the intersegment distance, which are the likely reasons of the FSC curves not dropping.

We carried forward all metadata of each segment during data processing, including the filament IDs and relative extraction coordinates along the helical axis (helical track length). After fixing a code error, as explained in the comment above, the FSC curves drop normally. We did not limit the x and y shift during alignment of the helical segments. The plot shown here shows the distribution of the shifts along the helical axis after alignment for the final reconstruction:

The mean shift was 0.03 Å with a standard deviation of 12.8 Å. The segment extraction spacing along the helical axis was 44.2 Å. This shows that, even though we did not limit the search range during alignment, the segments aligned locally at the extraction position.

For the local refinements (Extended Data Fig 2B), the authors used the symmetry expansion tool `relion_particle_symmetry_expand` with the option `--asu 3` (as stated in the method). Based on the extraction parameters, the “asu” option should have been 1, otherwise there is data duplication which can be an additional reason for the artifactual FSC curves (on top of the reasons mentioned for the helical refinement).

Data duplication within each half dataset should not inflate the FSC curves (and indeed they did drop to zero after we fixed our code error [see above]). But we have now calculated a new local reconstruction from the $[n_1=14, n_2=14]$ -symmetry stack with the option `--asu 1` (which corresponds to C14 symmetry expansion in this case here) as suggested by the reviewer.

The justifications on “high degree of symmetry, and the relatively small masked volume” should be removed, since those properties are present for many helical assemblies for which such artifacts are not present.

We have now removed this sentence from the manuscript. See comments above.

A.3/ The analysis of the nucleocapsid polymorphism by a supervised classification approach, although interesting and well thought -especially the reference preparation part-, suffer from severe drawbacks that make it difficult to take the author’s conclusions as granted.

First the authors should provide evidence of the correctness of the different helical symmetry assignments: this shouldn’t be difficult, since the classification provide them with different segment stacks and associated helical symmetry that could be used for refinement to provide high -or medium- resolution maps corresponding to the the different helical symmetries. On top of validating the classification procedure, this would allow the author to merge subtracted particles from different diameter tubes, hence improving the resolution of the local refinements and better characterizing potential lattice deformation (stretching, etc) by 3D classification or 3D variability analysis of the subtracted particles.

We have now calculated new helical and local reconstructions from a segment particle stack selected after supervised classification, which includes segments that classified to the dominant [$n_1=14$, $n_2=14$] symmetry (19,012 segments selected from the non-flattened and 1% and 2% flattened classes). Compared to the previously calculated helical reconstruction from 4,983 segments selected after 2D classification, we obtained an improved map as evident from the FSC analysis shown in Extended Data Fig. 2A, supporting our approach of selecting segments for a given symmetry by supervised classification. The resolution of the new local reconstruction is similar to the previous one, limited at 3.2 Å as shown by FSC analysis (Extended Data Fig. 2B), indicating that increasing the number of segments (and consequently the number of subparticles) did not further increase the resolution.

We also analyzed in more detail the 3D classification of the subparticle stack. As shown in an additional supplementary figure (Supplementary Fig. 7), we looked at the class partitioning of subparticles derived from either 0% or 2% flattened segments after classification without alignment of the subparticle stack. As one would expect, classes that displayed the largest radial shifts from the helical axis showed a substantial bias towards particles derived from the 2% flattened segments (classes 1, 7 and 8 in Supplementary Fig. 7) This analysis further supports the correct geometry assignment of segments by our supervised classification approach.

Performing refined 3D helical reconstructions for all geometries from the supervised classification would be computationally expensive and we believe it would not serve as an unbiased validation of the supervised classification result as one would again impose the same symmetry as used when the segments were initially classified. Thus, we think that such an analysis is beyond the scope of this work.

To provide evidence of the correct helical symmetries, we performed Fourier-Bessel analysis of the sum of the power spectra for the five most populated non-flattened classes and plotted the amplitude diagrams for the Bessel orders of the n_1 and n_2 diffraction peaks (Supplementary Fig. 6).

Another simple analysis that would strengthen the approach, would be to show the orientation distribution (on-axis and out-of-plane angles), for some representative references (e.g. the 5 most populated without flattening). The distribution is expected to be ~flat for on-axis angles, and gaussian-shaped, centered on 0 (or 90 depending on the convention) for the out-of-plane angles, if the symmetry assignment is correct. If the first proposed approach (calculating the 3D structures corresponding to the different helical symmetries) is beyond the scope of this work on author's point of view, then the authors should provide these plots for validation.

We plotted the distribution of the on-axis (rot), in-plane rotation angles (psi) and out-of-plane angles (tilt) for the five most populated, non-flattened classes (Revision Fig. 1). As expected, the distribution of the on-axis angles is flat, and the distribution of the out-of-plane angles is gaussian-shaped and centered on 90°. However, we do not believe that these distributions provide support for the correctness of the symmetry assignment. As one can see in the figure provided as attachment to this rebuttal (Revision Fig. 2), where we plotted for the 3,997 segments that classified as non-flattened [$n_1=14$, $n_2=14$]-symmetry the alignment angles when aligned to other (wrong) 3D references with symmetries of $n_1=12-16$ and $n_2=12-16$, respectively, the angle distributions look very similar and one wouldn't be able to tell which one is the correct symmetry. We therefore have not included such plots in the revised manuscript. Instead, we generated the average power spectra (after applying the psi angle and calculated the power spectrum) from segments of the five most populated, non-flattened classes and we

determined the Bessel orders of n_1 and n_2 in each power spectrum to verify the symmetry of these classes (Supplementary Fig. 6).

For the flattening, to be more conclusive, the authors could show the sum of power spectra of aligned segments corresponding to a particular helical symmetry (e.g. the most abundant), for the least and the most flattened classes. Effects of the flattening should be visible by the stretching of the signal on the layer lines.

This is an interesting suggestion and we have now added this analysis to the manuscript (Supplementary Movie 2).

Lines 345-347, the authors state that “A closer inspection of the segments belonging to tubes with $[n_1=14, n_2=14]$ -symmetry revealed that only approximately 14% of these segments aligned best to the non-flattened 3D reference.” Have the authors tried to run refinements before and after removing flattened segments, e.g. keeping up to flattening of 5%, and assess the improvements? On one hand, this would validate the classification based on flattening, and on the other hand integrate this classification procedure in their processing pipeline.

We have now calculated helical and local reconstructions from segments that classified with $[n_1=14, n_2=14]$ symmetry (up to and including 2% flattening). The helical reconstruction improved compared to the one that was calculated from a stack obtained after 2D classification as can be seen in the revised Extended Data Fig. 2A and C. We have updated the manuscript and Supplementary Fig. 1 to reflect integration of this protocol in the processing pipeline.

A.4/ Lines 572-587: the authors describe a complicated protocol for refined helical reconstruction, instead of simply using `relion_refine` with the helical options. Could the authors explain why? If the reason is the loss of metadata (priors on in-plane and out-of-plane, intersegment distance, filament IDs), then the authors should consider re-extracting in Relion using the filament ends coordinates corresponding to their subset of selected segments.

We appreciate the reviewers' questions regarding the complexity of our data processing approach and we are grateful for their call to explain our rationale for all steps during data processing better for clarity.

The complexity of this structure determination required the use of program tools from different software packages. We made sure that all metadata was carried forward when switching from one package to the other (including all the metadata associated with helical reconstructions, such as the filament ID, helical track length, and prior angles).

Briefly, we used `cisTEM` for 2D classification because its class averages featured more detailed subunits than Relion and Cryosparc. For 3D reconstruction of the two-times binned data, we first tried `cisTEM` because of how great its 2D classification algorithm worked for our segments. Tim Grant kindly shared an unpublished beta-version with us, but we were unable to generate a helical map. 3D reconstruction in Relion yielded maps with slightly flattened secondary structure features in the calculated maps. Helical reconstruction in `cryoSPARC` (called “helical refinement” in `cryoSPARC`) resulted in slightly better maps than Relion. Due to the large diameter of the segments, we were dealing with a large box size of 912 pixels for unbinned data. We used `cisTEM` for particle alignment when working with large box sizes (912 cubic voxels here) because of its speed and limitations due to memory requirements we frequently encountered when using other software packages for alignment. Alignment, reconstruction, and symmetrization are just treated as separate steps in our protocol. In addition, subparticle

extraction was essential for local refinement. Unfortunately, subparticle extraction from a movie-refine corrected stack is not possible with Relion (only from the original summed images). For the supervised classification approach, we used cisTEM for the following reasons: Global alignment of the 72K segments was very slow in Relion. In addition, we relied heavily on parallelization, which was easier with cisTEM than we Relion or cryoSPARC. At one point, we used 6,000 CPUs simultaneously. Furthermore, helical refinement in cryoSPARC does not (yet?) offer a local symmetry implementation.

Minor points:

B.1/ Lines 144-149: “We initially used ModelAngelo [...] in Phenix”. Most of this paragraph seem more appropriate for the method section.

We agree that the sentence “Real-space refinement and model validation were performed in Phenix.” in lines 148-149 is more appropriate for the Methods section and we deleted it from this paragraph. However, we think that reporting the relatively high success rate of automated model building by ModelAngelo (64% sequence identity, 77% sequence similarity) is appropriate for the Results section because it confirms the quality of the reconstruction as opposed to the biased eyes of the human model builder. We now refer to Supplementary Table 1 at the end of the paragraph.

The paragraph was updated to:

“We initially used ModelAngelo for model building⁴⁵, where we observed that—without providing an amino-acid sequence—the program was able to output an almost complete trace of the VP39 structure with a high degree of correct amino acid assignments (64% identity, 77% similarity), confirming the visually assessed quality of the reconstruction. We completed the model by manual building in Coot⁴⁶. The first 11 residues of the N terminus and the last 27 residues of the C terminus of VP39 were unresolved in our cryo-EM reconstruction and were not included in the model (Supplementary Table 1).”

B.2/ Lines 132-133: “Our reconstruction reveals that individual VP39 subunits pack as dimers that assemble into 14 protofilaments”. This is a general statement that does not apply to the entire dataset, it should be more precisely phrased for the readers to understand that it also assembles in other number of protofilaments structures.

We updated the text in lines 132-134 to “**In our reconstruction**, individual VP39 subunits pack as dimers that assemble into **14 helical strands** and together form the central cylindrical structure of the baculoviral nucleocapsid (Fig. 1C).” We hope that this sentence clarifies that the 14 helical strands are only observed in the reconstruction that is described in this section of the manuscript.

B.3/ Line 532: “with a search range of -15° to 15° , rise: 43.8 Å with a search range of 20-60 Å, C15”. This should be C14.

We thank the reviewer for highlighting this description as unclear. This is not a typo; C15 is correct here. This dataset is unusually complex in terms of its size and the sample heterogeneity. This complexity made the determination of helical symmetry parameters and the helical reconstruction inherently challenging. When we started with 3D reconstruction, we were not certain yet about the correct rotational symmetry of the segments. At this point, we knew from helical indexing of early 2D class averages that the rotational symmetry is likely C12-C15. The C15 map, which we used as an input model at this step during data processing, looked

more detailed than a C14 map that we generated simultaneously in a brute-force approach using the same segments, where we tested for different possible rotational symmetries. This may be due to the fact that the C15 map contained more segments with C15-symmetry than maps, which we generated later. Our selection of 2D classes, which was used to generate the C15 map, was not that precise yet. We did not strictly adhere to the exact same segment diameter for picking 2D classes yet. Instead, we picked all classes that looked alike but may have differed slightly in tube diameter. Latter maps contain fewer segments due to finer selection during 2D classification. However, the C15 map was good enough as a first input model.

We added the following sentence to the manuscript in line 536 for clarification: **“Finer selection in these and subsequent rounds of 2D classification, where we focused exclusively on class averages with C14 rotational symmetry, led to switching from C15 to C14 symmetry for helical reconstruction.”**

B.4/ Supplementary Fig 2A: “Power spectrum of a 2D class average, which has been symmetrized along its meridian”. Instead of the PS of the 2D class average, it is recommended to use the sum of PS of the segments belonging to that class-average, which avoid some artifacts present in the PS of class-averages, such as left/right asymmetry. No symmetrization should be necessary if the PS is not artifactual.

We thank the reviewer for this comment. We have now calculated the sum of the power spectra of the segments, repeated the helical indexing, and we have now updated Supplementary Fig. 2 in the manuscript with the new power spectra and curves. As the reviewer suggested, no symmetrization was necessary.

B.5/ Lines 315-316: “potential symmetries (C12-C15) possible for the given Bessel orders identified during Fourier-Bessel analysis of several 2D class averages (Supplementary Fig. 2).” Supp Fig 2 shows only the analysis on one class, therefore the reader can not appreciate the validity of this statement and the differences in first Bessel peak position across the different structures (which might actually be compensated by diameter variation).

We thank the reviewer for pointing out the benefit of showing the observation of other potential symmetries. We have removed this sentence from the manuscript. These 2D class averages were obtained prior to finer selection during 2D classification and thus may include closely related symmetries in one class average (also see comment B.3). Deducing the correct symmetry at this step is very challenging.

B.6/ Lines 544-545 : “We determined initial helical symmetry parameters by indexing the power spectra of these class averages using PyHI (Python v. 3.7)”. It is not clear why this indexation was done, since the authors already obtained helical parameters in previous steps.

We appreciate the reviewer’s thoroughness and efforts to help us clarify our data processing approach. The scheme in Supplementary Fig. 1 shows the full final data processing approach for the data collections, which resulted in this reconstruction. In this paper, we only discuss data, which we collected on graphene-coated grids. However, prior to using graphene-coated grids, we tried to reconstruct the nucleocapsid with data that we collected from conventional carbon support grids (Quantifoil). The helical parameters, which we used for the extraction of the segments described in this paper, stem from helical indexing of 2D class averages of data collected from the conventional carbon grids. The nucleocapsids for all data collections were prepared using the same virus and same purification protocol.

While we collected a similar number of movies, we observed about five times less tubes on the conventional Quantifoil grids, compared to the graphene-coated grids (16,402 segments were extracted from 45,000 micrographs for conventional Quantifoil grids; 74,600 segments were extracted from 46,000 micrographs of graphene-coated gold grids). The 16K segments were not sufficient to get beyond a blobby low-resolution structure and we were not able to confirm the helical symmetry parameters.

We updated the Results section in line 116 to: “Cryo-EM data collection was greatly facilitated by graphene-supported grids, which increased particle yield per micrograph by a factor of 5 **and made this reconstruction possible**⁴². **With an initial data set collected from conventional Quantifoil grids (16,402 segments from 45,000 micrographs), we were only able to obtain a low-resolution reconstruction.**

In addition, we added an explanation to the Methods section in line 526:

“The helical symmetry parameters, which were used for segment extraction, were obtained from Fourier-Bessel analysis of 2D class averages of previously collected data of AcMNPV VP39 nucleocapsids on conventional Quantifoil grids (as opposed to graphene-coated gold grids).”

B.7/ For the local refinements, it should be stated whether refinement was done with C2 (or D1) symmetry, or only in C1. Since the symmetry expansion was done with C14 and not D14, we would recommend the authors to use the 2-fold symmetry for refinement.

We carried out the local refinement with C1 setting, but applied local symmetry after each iteration, thereby averaging the density of all protomers (8) that were included in the volume. Thus the 2-fold symmetry was used for refinement. We have now written this more clearly in the Methods section.

B.8/ Validation report, section 6.2.1/6.2.2: While the “raw map” seem to have 2-fold cyclic symmetry, or at least have one of its axis (X) aligned with a (maybe imposed, see point B.7 to clarify) C2 symmetry axis, this is not the case for the “primary map”. Why such a difference?

The “primary” and “raw” maps were the same, except that the “primary” map was boxed.

In the “raw map” central slices, can the author explain the very strong differences in areas with no protein density near the center and the areas with no protein density far from the center? In other words, why is the noise appearance so different depending on the area of the volume? It looks like a different low-pass filter had been applied in different areas, which seem odd for a “raw map”.

Good observation. The “raw” map was not an unmodified map (as it should be) and was local symmetry averaged. The noise in the regions where the local symmetry mask was applied (around each protomer) is much lower than on the outside. We have now revised the deposition and uploaded all relevant maps (see also comment above).

Revision Fig. 1

Out-of-plane angles

In-plane rotation angles

On-axis angles

Revision Fig. 2

REVIEWER COMMENTS

Reviewer #1 (Remarks to the Author):

The authors have done a very good job in addressing my concerns, and in general the paper is now suitable for publication. However, I would take exception to the argument made here in responding to Reviewer #2 and the inclusion of Supp. Fig. 6 as establishing the correctness of their helical indexing:

"To provide evidence of the correct helical symmetries, we performed Fourier-Bessel analysis of the sum of the power spectra for the five most populated non-flattened classes and plotted the amplitude diagrams for the Bessel orders of the n_1 and n_2 diffraction peaks (Supplementary Fig. 6)."

The agreement between the layer line intensities of the projected 3D reconstruction and that from the averaged power spectrum of the raw segments is a necessary, but not sufficient, condition for having used the correct symmetry. This is actually shown in a Methods in Enzymology chapter (Egelman, 2010) where the degeneracy of symmetries is shown and discussed, illustrating how wrong symmetries (at some finite resolution) can be indistinguishable in terms of layer line intensities. It is stated in the legend for Fig. 6.8 in that chapter: "Despite having different Bessel orders on layer lines 1, 2, 4, and 5, the power spectra all have peaks at identical positions due to the fact that the diffraction is coming from different radii." At some higher resolution, one would see a divergence between the reconstruction and the raw data, but this resolution might be significantly beyond what is present in the power spectrum from the raw segments. Thus, the main argument for having used the correct symmetry is producing an interpretable map at high resolution, and that one symmetry (the correct one) produces a better map than all other symmetries possible. An example was published (Zheng et al., 2020) where at 5 Å resolution two different symmetries produced indistinguishable volumes, and thus would have had indistinguishable power spectra. However, the correct symmetry led to a 3.9 Å map, while the incorrect symmetry never improved beyond 5 Å.

Egelman, E.H. (2010). Reconstruction of helical filaments and tubes. *Meth. Enzymol.* 482, 167-183.

Zheng, W., Pena, A., Low, W.W., Wong, J.L.C., Frankel, G., and Egelman, E.H. (2020). Cryoelectron-Microscopic Structure of the pKpQIL Conjugative Pili from Carbapenem-Resistant *Klebsiella pneumoniae*. *Structure* 28, 1321-1328 e1322.

Reviewer #2 (Remarks to the Author):

In the reviewed version of their article "Helical reconstruction of VP39 baculovirus nucleocapsid assembly reveals principles for baculovirus nucleocapsid assembly", the authors have addressed a

number of points of concern, including technical errors that lead to the inflation in their FSC curves, performed new validations of their classification procedure, as well as slightly adapted some steps of their processing strategy. The authors have added new supplementary figures and corrected their text based on reviewers comments.

While this improved manuscript seem suitable for publication (with a minor correction proposed), and the detailed explanations were highly appreciated, some of the point-to-point answers require comments, as followed.

Minor correction : the middle panels in supplementary fig. 7 (AnglePsi) do not seem to add anything to the analysis of the classification, and is proposed to be removed. Indeed, those angles are random and not linked to any geometrical considerations, unlike the AngleRot or AngleTilt.

Comments on the point-to-point answers :

“Performing refined 3D helical reconstructions for all geometries from the supervised classification would be computationally expensive and we believe it would not serve as an unbiased validation of the supervised classification result as one would again impose the same symmetry as used when the segments were initially classified. Thus, we think that such an analysis is beyond the scope of this work.”

The authors have performed additional helical and local reconstructions for the 14,14 symmetry (which contain the most segments and hence will take longer to compute) in the reviewing process, the argument of the processing time seems therefore unjustified. The fact that the same symmetry as used for the classification procedure is imposed is not a problem : the validation would come from the fact that the obtained reconstructions are correct and at high resolution (which would be only true if the correct symmetry has been imposed).

“We plotted the distribution of the on-axis (rot), in-plane rotation angles (psi) and out-of-plane angles (tilt) for the five most populated, non-flattened classes (Revision Fig. 1). As expected, the distribution of the on-axis angles is flat, and the distribution of the out-of-plane angles is gaussian-shaped and centered on 90.”

Unlike stated, the distribution of the on-axis angles is only flat for the 14,14 map, but not for the 13,14 (although nearly ok) and really not for the 15,14 map, while the other symmetries have too few segments to judge. Together with the not always well fitting data/predicted PS as shown in the new supplementary figure 6, and with the fact that the authors have not calculated helical reconstruction for those other symmetries, there is a remaining doubt about the correctness of the symmetry assignments based on the reference based classification approach.

“As one can see in the figure provided as attachment to this rebuttal (Revision Fig. 2), where we plotted for the 3,997 segments that classified as non-flattened [n1=14, n2=14]-symmetry the alignment angles when aligned to other (wrong) 3D references with symmetries of n1=12–16 and n2=12–16, respectively, the angle distributions look very similar and one wouldn't be able to tell which one is the correct symmetry.”

For the AngleRot plots (which are the most interesting ones, when it comes to detecting wrong symmetry imposed), the fact that the distribution is not even when the correct symmetry is imposed (unlike what is shown in Revision Fig. 1, why ?) is a problem and may indicate wrong translational search range parameters (which should be limited to +/- half the axial rise in Y). Indeed, in those conditions, it is hard to judge the correctness of the maps from the AngleRot plots, but the authors should be aware that these are often useful indications (and should maybe make them worry about the 15,14 map plots shown in Revision Fig 1).

EDITOR COMMENTS

You will see that, while the reviewers find that your revisions improved the manuscript, some important points remain to be addressed. Please ensure all requests are undertaken including calculation of helical reconstructions for the symmetries that they are excluding as possible fits and the inclusion of a Supplementary Figure to further explore this area.

REVIEWER COMMENTS

Reviewer #1 (Remarks to the Author):

The authors have done a very good job in addressing my concerns, and in general the paper is now suitable for publication. However, I would take exception to the argument made here in responding to Reviewer #2 and the inclusion of Supp. Fig. 6 as establishing the correctness of their helical indexing:

"To provide evidence of the correct helical symmetries, we performed Fourier-Bessel analysis of the sum of the power spectra for the five most populated non-flattened classes and plotted the amplitude diagrams for the Bessel orders of the n1 and n2 diffraction peaks (Supplementary Fig. 6)."

The agreement between the layer line intensities of the projected 3D reconstruction and that from the averaged power spectrum of the raw segments is a necessary, but not sufficient, condition for having used the correct symmetry. This is actually shown in a Methods in Enzymology chapter (Egelman, 2010) where the degeneracy of symmetries is shown and discussed, illustrating how wrong symmetries (at some finite resolution) can be indistinguishable in terms of layer line intensities. It is stated in the legend for Fig. 6.8 in that chapter: "Despite having different Bessel orders on layer lines 1, 2, 4, and 5, the power spectra all have peaks at identical positions due to the fact that the diffraction is coming from different radii." At some higher resolution, one would see a divergence between the reconstruction and the raw data, but this resolution might be significantly beyond what is present in the power spectrum from the raw segments. Thus, the main argument for having used the correct symmetry is producing an interpretable map at high resolution, and that one symmetry (the correct one) produces a better map than all other symmetries possible. An example was published (Zheng et al., 2020) where at 5 Å resolution two different symmetries produced indistinguishable volumes, and thus would have had indistinguishable power spectra. However, the correct symmetry led to a 3.9 Å map, while the incorrect symmetry never improved beyond 5 Å.

Egelman, E.H. (2010). Reconstruction of helical filaments and tubes. *Meth. Enzymol.* 482, 167-183.

Zheng, W., Pena, A., Low, W.W., Wong, J.L.C., Frankel, G., and Egelman, E.H. (2020). Cryoelectron-Microscopic Structure of the pKpQIL Conjugative Pili from Carbapenem-Resistant *Klebsiella pneumoniae*. *Structure* 28, 1321-1328 e1322.

We agree with the Reviewer that an interpretable map at high resolution is the main argument to confirm correctness of the symmetry assignment, as we have done for our main [14,14]-symmetric reconstruction (3.6 Å resolution of the helical reconstruction). Indeed, our analysis produced a map where ModelAngelo was able to build an almost complete model, with a high degree of confidence in placing the correct amino acids without providing an amino acid sequence (as described in lines 153-157). The final fit of our model, with correct stereochemistry and validated, fits into the map as one would expect.

We have now calculated reconstructions from segments that partition into classes with helical symmetries other than [14,14] after supervised classification, as summarized in Supplementary Fig. 8 and Supplementary Table 5. See comment below to Reviewer #2.

We agree with the Reviewer that “the agreement between the layer line intensities of the projected 3D reconstruction and that from the averaged power spectrum” may not be a sufficient condition for having used the correct symmetry. In fact, the correct interpretation of the power spectra from these assemblies is extremely complicated (e.g., if the helical repeat distance is very large, the selection rule becomes complicated, resulting in many layer lines). In order to not rely on Fourier-Bessel indexing, we have completely revised Supplementary Fig. 6. It now shows directly the fit between the power spectra of the five most populated, non-flattened classes and back-projections of the 3D references they classified with, without relying on Fourier-Bessel indexing. In panel A, we show the averaged power spectra of the segments after applying the psi alignment angle. In panel B, we show the averaged power spectra calculated from reference projections after orienting them with the same alignment parameters as the corresponding observed segments. Panel C shows the layer line profiles for prominent layer lines (observed vs calculated) for each class without indexing. We also made Supplementary Movie 3 (comparison between observed and calculated power spectra) and Supplementary Movie 4 (comparison of the power spectra from different symmetries). We also added an extra sentence in line 359 in the results section:

“The imposed symmetries for segments partitioning into 3D references are consistent when comparing the direct fit of averaged power spectra of calculated and observed segments (Supplementary Fig. 6, Supplementary Movies 3 and 4). Given the caveats associated with analyzing Fourier spectra^{57,58}, interpretable maps would be the best way to verify correct helical symmetry assignments. However, while we calculated reconstructions for segments of the three most populated, 0-2% flattened classes (Supplementary Fig. 8), we did not have a sufficiently high number of segments to yield an interpretable map to unambiguously prove the correct helical symmetry assignments for symmetries other than our [14,14] reconstruction (Supplementary Table 5).”

(57) Egelman, E.H. Reconstruction of helical filaments and tubes. *Methods Enzymol* **482**, 167-83 (2010).

(58) Zheng, W. et al. Cryoelectron-Microscopic Structure of the pKpQIL Conjugative Pili from Carbapenem-Resistant *Klebsiella pneumoniae*. *Structure* **28**, 1321-1328.e2 (2020).

Reviewer #2 (Remarks to the Author):

In the reviewed version of their article “Helical reconstruction of VP39 baculovirus nucleocapsid assembly reveals principles for baculovirus nucleocapsid assembly”, the authors have addressed a number of points of concern, including technical errors that lead to the inflation in their FSC curves, performed new validations of their classification procedure, as well as slightly adapted some steps of their processing strategy. The authors have added new supplementary figures and corrected their text based on reviewers comments.

While this improved manuscript seem suitable for publication (with a minor correction proposed), and the detailed explanations were highly appreciated, some of the point-to-point answers require comments, as followed.

Minor correction: the middle panels in supplementary fig. 7 (AnglePsi) do not seem to add anything to the analysis of the classification, and is proposed to be removed. Indeed, those angles are random and not linked to any geometrical considerations, unlike the AngleRot or AngleTilt.

We have revised the old Supplementary Fig. 7 as suggested by removing the AnglePsi distribution plots.

Comments on the point-to-point answers:

“Performing refined 3D helical reconstructions for all geometries from the supervised classification would be computationally expensive and we believe it would not serve as an unbiased validation of the supervised classification result as one would again impose the same symmetry as used when the segments were initially classified. Thus, we think that such an analysis is beyond the scope of this work.”

The authors have performed additional helical and local reconstructions for the 14,14 symmetry (which contain the most segments and hence will take longer to compute) in the reviewing process, the argument of the processing time seems therefore unjustified. The fact that the same symmetry as used for the classification procedure is imposed is not a problem: the validation would come from the fact that the obtained reconstructions are correct and at high resolution (which would be only true if the correct symmetry has been imposed).

The main difference in processing time between calculating a [14,14]-symmetric reconstruction, and one without rotational symmetry, e.g. [13,14], stems from the fact that the later requires map symmetrization by applying local symmetry (which is slower) instead of applying point-group symmetry. We have now implemented local symmetry calculation for map symmetrization and calculated helical reconstructions for the most populated classes after supervised classification. The result is summarized in a new Supplementary Table 5. It shows that the resolution obtained for the [13,14] and [15,14] symmetries is comparable to the resolution obtained for a [14,14] reconstruction calculated from a similar number of segments, and thus consistent. But one can also see that limiting the number of segments to about 1200 does not yield high-resolution reconstructions, even for the [14,14] class. Therefore, we do not have a sufficient number of segments to unambiguously prove the correctness of the symmetry assignment for the other classes. We have added a sentence in the results section explaining this limitation in line 359 and a description in the methods section starting in line 693.

“We plotted the distribution of the on-axis (rot), in-plane rotation angles (psi) and out-of-plane angles (tilt) for the five most populated, non-flattened classes (Revision Fig. 1). As expected, the distribution of the on-axis angles is flat, and the distribution of the out-of-plane angles is gaussian-shaped and centered on 90°.”

Unlike stated, the distribution of the on-axis angles is only flat for the 14,14 map, but not for the 13,14 (although nearly ok) and really not for the 15,14 map, while the other symmetries have too few segments to judge. Together with the not always well fitting data/predicted PS as shown in the new supplementary figure 6, and with the fact that the authors have not calculated helical reconstruction for those other symmetries, there is a remaining doubt about the correctness of the symmetry assignments based on the reference based classification approach.

We have revised Revision Fig. 1 and 2, where we correctly account now for the redundancy of AngRot angles for any given helical symmetry. The distributions of the AngRot angles are flat for all helical symmetries in Revision Fig. 1 and 2 (where there are enough segments to obtain a

representative distribution). Please see response to last comment for explanation and revision of the figures.

We have completely revised Supplementary Fig. 6 in order to show directly the fit between the power spectra without relying on Fourier-Bessel indexing by the PyHI program. The analysis is still for the same five most populated helical symmetries. In panel A, we show the averaged power spectra of the segments after rotating the segments according to the ψ angle from the 3D alignment. In panel B, we show the averaged power spectra calculated from reference projections after orienting them with the same alignment parameters as observed for each segment and applying the ψ alignment angle. Panel C shows the layer line profiles for prominent layer lines (observed vs calculated) for each helical symmetry. We have also added Supplementary Movie 3 that shows the fit between the observed and calculated averaged power spectra for the five most populated helical symmetries, and Supplementary Movie 4 that shows the difference between the power spectra from different helical symmetries. As can be seen in Supplementary Fig. 6 and Supplementary Movie 3, the observed and calculated power spectra agree well, supporting the correctness of the supervised classification result.

“As one can see in the figure provided as attachment to this rebuttal (Revision Fig. 2), where we plotted for the 3,997 segments that classified as non-flattened [$n1=14$, $n2=14$]-symmetry the alignment angles when aligned to other (wrong) 3D references with symmetries of $n1=12-16$ and $n2=12-16$, respectively, the angle distributions look very similar and one wouldn't be able to tell which one is the correct symmetry.”

For the AngleRot plots (which are the most interesting ones, when it comes to detecting wrong symmetry imposed), the fact that the distribution is not even when the correct symmetry is imposed (unlike what is shown in Revision Fig. 1, why ?) is a problem and may indicate wrong translational search range parameters (which should be limited to \pm half the axial rise in Y). Indeed, in those conditions, it is hard to judge the correctness of the maps from the AngleRot plots, but the authors should be aware that these are often useful indications (and should maybe make them worry about the 15,14 map plots shown in Revision Fig 1).

We have revised Revision Fig. 1 and 2 to represent the true unbiased AngleRot distribution. For the AngleRot plots, the distributions are in fact even or random (as one, including Reviewer #2, would expect). Because of helical symmetry, there are redundant and equivalent (e.g. giving an identical alignment score) alignment angles and shifts for each segment, when the alignment is carried out in C1 (as we did) against non-flattened references. Which one of these equivalent solutions the alignment program reports might be random or not (as it appears to be the case of cisTEM used here).

In order to represent the true unbiased AngleRot distribution, we must account for the redundancy of AngleRot angles, depending on the helical symmetry, between 0 and 360 degrees before plotting the distributions. For this, we used the following python code before plotting the distributions, where *twist1* and *twist2* are the helical twists associated with the symmetry defined by *n1* and *n2*:

```
#####  
# python code to account for redundant AngleRot angles after C1 alignment  
  
n1 = 15  
n2 = 14  
  
twist1 = np.abs(twist1)
```

```

twist2 = np.abs(twist2)

if (n1 != n2):      # no rotation symmetry
    twist_rot = 360.0
else:              # rotational symmetry
    twist_rot = 360.0/n1

q1 = np.floor(360.0/twist1)
q2 = np.floor(360.0/twist2)
q3 = np.floor(360.0/twist_rot)

rot = [r
+np.random.randint(0,q1+1)*twist1
+np.random.randint(0,q2+1)*twist2
+np.random.randint(0,q3+1)*twist_rot
for r in AngleRot]

for i in range(3):
    rot = [r if r<360.0 else r-360.0 for r in rot]
#####

```

The revised distribution plots are shown in Revision Fig. 1_revision_02 and 2_revision_02. As one can see, the distributions look even for all symmetries. Thus, we believe that there is no indication for wrong translational search range parameters or incorrectness of the maps.

We have updated Supplementary Fig. 7 by applying the above code before plotting the AngleRot distributions.

In the previous revision, we only accounted for this with the rotational symmetry in case of the [14,14] assembly in Revision Fig. 1, but not in the plots of Revision Fig. 2, which is why they looked different.

REVIEWER COMMENTS

Reviewer #1 (Remarks to the Author):

The authors have done a very thorough job in responding to the points raised by the reviewers. I think that the changes that have been made, along with the caveats now inserted into the manuscript, make the present version suitable for publication.

I would just like to raise one issue in response to an argument made in the Response:

"In order to not rely on Fourier- Bessel indexing, we have completely revised Supplementary Fig. 6. It now shows directly the fit between the power spectra of the five most populated, non-flattened classes and back-projections of the 3D references they classified with, without relying on Fourier-Bessel indexing."

In fact, whether one uses Fourier-Bessel indexing or compares directly the fit between the observed power spectra from the real segments and the power spectrum of the 3D symmetrized volume, makes absolutely no difference. In both cases, one can get a good match with the wrong symmetry. That is why this match is a necessary, but not sufficient, criterion for having used the correct symmetry. Nothing needs to be changed in the paper or Supplement, but I thought it helpful to raise this point.

Reviewer #2 (Remarks to the Author):

In the revised version of their article "Helical reconstruction of VP39 baculovirus nucleocapsid assembly reveals principles for baculovirus nucleocapsid assembly", the authors have tried to address the reviewers concerns about the correctness of some helical symmetry assignments, by the revision of supplementary figure 6 and the addition of a new supplementary figure 8. Although I still support the publication of this interesting work, the new figures now highlights issues that are related to concerns raised from the first reviewing round, that now appear clearer, and require major revisions before publication, as detailed below.

A/ Symmetry concerns

In Supplementary fig. 6 and 8, the authors now show more details on the reconstructions obtained for other symmetries than the main [14,14]-symmetry after their supervised classification approach. In the panel A of supp fig 6, the averaged power spectra of segments corresponding to the [15,14] and the [16,16] symmetries appear very asymmetrical (left/right asymmetry), in particular for layer line labelled "1", and the layer line below "2".

Power spectra of 2D projections of helical structures are mirror symmetric by definition, and the departure from this symmetry is indicative of a serious issue. If we rule out the possibility that their filaments do not possess helical symmetry, and possible air-water interface denaturation that would also break helical symmetry, the most likely explanation is related to alignment issues (in this case, the

alignment of the segments to 2D projections of 3D reference structures used during the classification procedure). Such misalignment typically appears when the reference structure has a different helical symmetry than the symmetry of the segments that are aligned to it. Therefore, the supp fig 6 now almost proves that the assignment of segments to [15,14] and [16,16] symmetries is wrong. Among those two symmetries, the supp fig 8 shows only the [15,14] symmetry map, and this map shows different features than the two other maps ([14,14]-which is the ground truth- and [13,14]) : lateral and vertical interactions appear different, and the most external long alpha helix does not fit into the map at all.

In brief, the [14,14] and the [13,14] maps show comparable and apparently reliable densities and show symmetric PS averages, while the [15,14] map shows apparently unreliable densities and asymmetric PS averages : the simplest conclusion is that while the former are correct, the latter is not.

The authors have two alternatives to deal with this issue. The first option is to keep the figures, including the [15,14] and [16,16] asymmetric PS average in supp fig 6 panel A and the corresponding [15,14] map in supp fig 8, but to label them as “possibly wrong” symmetry assignment both on the figures (e.g. a red cross on the PS) and in the text. While this first option would serve educational purposes, the second option is to simply remove anything related to the [15,14] and [16,16] symmetries. In any case, it is not possible to show such PS averages as being “normal”, this would be misleading for readers and newcomers to the field of helical processing.

B/ Supervised classification

It should now become clear to the authors that their proposed supervised classification approach, although partially valid (e.g. the [13,14] symmetry assignment seems correct), can also lead to wrong symmetry assignments. Therefore, it is proposed that a warning is given, stating the possibility that not all symmetry assignment are correct. This method is not be broadly applicable to symmetry determination in cases similar to the one presented in this article, and will likely always give a high rate of “false positives” (the explanations for that go beyond the scope of this review).

C/ Minor revision

-Text in Supp Fig 6 panel C is not readable, font size too small

REVIEWER COMMENTS

Reviewer #1 (Remarks to the Author):

The authors have done a very thorough job in responding to the points raised by the reviewers. I think that the changes that have been made, along with the caveats now inserted into the manuscript, make the present version suitable for publication.

We truly appreciate the time and effort of Reviewer 1 in carefully evaluating the manuscript and helping address the remaining comments.

I would just like to raise one issue in response to an argument made in the Response:

"In order to not rely on Fourier- Bessel indexing, we have completely revised Supplementary Fig. 6. It now shows directly the fit between the power spectra of the five most populated, non-flattened classes and back-projections of the 3D references they classified with, without relying on Fourier-Bessel indexing."

In fact, whether one uses Fourier-Bessel indexing or compares directly the fit between the observed power spectra from the real segments and the power spectrum of the 3D symmetrized volume, makes absolutely no difference. In both cases, one can get a good match with the wrong symmetry. That is why this match is a necessary, but not sufficient, criterion for having used the correct symmetry. Nothing needs to be changed in the paper or Supplement, but I thought it helpful to raise this point.

We fully agree that the match is necessary but not sufficient. We appreciate raising this point.

Reviewer #2 (Remarks to the Author):

In the revised version of their article "Helical reconstruction of VP39 baculovirus nucleocapsid assembly reveals principles for baculovirus nucleocapsid assembly", the authors have tried to address the reviewers concerns about the correctness of some helical symmetry assignments, by the revision of supplementary figure 6 and the addition of a new supplementary figure 8. Although I still support the publication of this interesting work, the new figures now highlights issues that are related to concerns raised from the first reviewing round, that now appear clearer, and require major revisions before publication, as detailed below.

A/ Symmetry concerns

In Supplementary fig. 6 and 8, the authors now show more details on the reconstructions obtained for other symmetries than the main [14,14]-symmetry after their supervised classification approach. In the panel A of supp fig 6, the averaged power spectra of segments corresponding to the [15,14] and the [16,16] symmetries appear very asymmetrical (left/right asymmetry), in particular for layer line labelled "1", and the layer line below "2".

Power spectra of 2D projections of helical structures are mirror symmetric by definition, and the departure from this symmetry is indicative of a serious issue. If we rule out the possibility that their filaments do not possess helical symmetry, and possible air-water interface denaturation that would also break helical symmetry, the most likely explanation is related to alignment issues (in this case, the alignment of the segments to 2D projections of 3D reference structures used during the classification procedure). Such misalignment typically appears when the reference structure

has a different helical symmetry than the symmetry of the segments that are aligned to it. Therefore, the supp fig 6 now almost proves that the assignment of segments to [15,14] and [16,16] symmetries is wrong. Among those two symmetries, the supp fig 8 shows only the [15,14] symmetry map, and this map shows different features than the two other maps ([14,14]-which is the ground truth- and [13,14]) : lateral and vertical interactions appear different, and the most external long alpha helix does not fit into the map at all.

In brief, the [14,14] and the [13,14] maps show comparable and apparently reliable densities and show symmetric PS averages, while the [15,14] map shows apparently unreliable densities and asymmetric PS averages : the simplest conclusion is that while the former are correct, the latter is not.

The authors have two alternatives to deal with this issue. The first option is to keep the figures, including the [15,14] and [16,16] asymmetric PS average in supp fig 6 panel A and the corresponding [15,14] map in supp fig 8, but to label them as “possibly wrong” symmetry assignment both on the figures (e.g. a red cross on the PS) and in the text. While this first option would serve educational purposes, the second option is to simply remove anything related to the [15,14] and [16,16] symmetries. In any case, it is not possible to show such PS averages as being “normal”, this would be misleading for readers and newcomers to the field of helical processing.

We have taken the second option and removed anything related to the [15,14]- and [16,16]-symmetries in Figure 6 and Supplemental Figure 8, and thank Reviewer 2 for a clear path to resolving this concern.

Also, in response, we offer the following comments regarding asymmetric power spectra: First, we suspect it is very likely that there may have been air-water interface damage that is asymmetric, given the large diameter of the tubes. Even excluding this likely consideration, we note that if the tube segments are tilted in plane due to misalignment, the layer line at zero should exhibit a fan-shaped fuzzy pattern. Since all power spectra are comparable to the [14,14] power spectra, we suspect incorrect alignment to be unlikely. Further, if the number of tubes used to generate the average power spectra are not large, an uneven of distribution of tube segments sticking to the top or bottom air-water interface may result in asymmetry.

We only offer these comments for completeness, acknowledge and thank Reviewer 2 for bringing this issue to our attention, and have taken the simplest suggested course of action and removed the related panels.

B/ Supervised classification

It should now become clear to the authors that their proposed supervised classification approach, although partially valid (e.g. the [13,14] symmetry assignment seems correct), can also lead to wrong symmetry assignments. Therefore, it is proposed that a warning is given, stating the possibility that not all symmetry assignment are correct. This method is not be broadly applicable to symmetry determination in cases similar to the one presented in this article, and will likely always give a high rate of “false positives” (the explanations for that go beyond the scope of this review).

We have added a line warning that:

“We note that caution should be taken when applying this supervised classification approach, as it may generate incorrect symmetry assignments along with assignments that are correct. The results of our supervised classification require careful analysis.”

C/ Minor revision

-Text in Supp Fig 6 panel C is not readable, font size too small

This has been corrected.

REVIEWERS' COMMENTS

Reviewer #2 (Remarks to the Author):

The authors have included suggested changes and the manuscript is now suitable for publication. As a side note, I'd like to emphasize that the comments were not meant to be interpreted as if the [15,14] and [16,16] symmetries do not exist in the dataset. They most likely exist, but as judged from the PS sums the classification / alignment were not suitable for correct structure determination.

REVIEWERS' COMMENTS

Reviewer #2 (Remarks to the Author):

The authors have included suggested changes and the manuscript is now suitable for publication.

As a side note, I'd like to emphasize that the comments were not meant to be interpreted as if the [15,14] and [16,16] symmetries do not exist in the dataset. They most likely exist, but as judged from the PS sums the classification / alignment were not suitable for correct structure determination.

We appreciate Reviewer #2's comment and agree.